# A non-genetic, cell cycle-dependent mechanism of platinum resistance in lung adenocarcinoma

Alvaro Gonzalez Rajal[1,2,3]*, Kamila A Marzec[1], Rachael A McCloy[4], Max Nobis[3,4], Venessa Chin[3,4,5], Jordan F Hastings[4], Kaitao Lai[1,6], Marina Kennerson[1,6], William E Hughes[2,3,7], Vijesh Vaghjiani[8], Paul Timpson[3,4], Jason E Cain[8,9], D Neil Watkins[10,11], David R Croucher[3,4†]*, Andrew Burgess[1†]*

[1]ANZAC Research Institute, Concord Hospital, Concord, Australia; [2]Garvan Institute of Medical Research, Sydney, Australia; [3]St Vincent's Hospital Clinical School, University of New South Wales, Sydney, Australia; [4]The Kinghorn Cancer Centre, Garvan Institute of Medical Research, Sydney, Australia; [5]St Vincent's Hospital Sydney, Darlinghurst, Australia; [6]The University of Sydney Concord Clinical School, Faculty of Medicine and Health, Sydney, Australia; [7]Children's Medical Research Institute, The University of Sydney, Westmead, Australia; [8]Hudson Institute of Medical Research, Clayton, Australia; [9]Department of Molecular and Translational Medicine, School of Medicine, Nursing and Health Sciences, Monash University, Clayton, Australia; [10]Research Institute in Oncology and Hematology, CancerCare Manitoba, Winnipeg, Canada; [11]Department of Internal Medicine, Rady Faculty of Health Science, University of Manitoba, Winnipeg, Canada

**\*For correspondence:**
a.rajal@garvan.org.au (AGR);
d.croucher@garvan.org.au (DRC);
andrew.burgess@sydney.edu.au (AB)

†These authors contributed equally to this work

**Competing interests:** The authors declare that no competing interests exist.

**Abstract** We previously used a pulse-based in vitro assay to unveil targetable signalling pathways associated with innate cisplatin resistance in lung adenocarcinoma (Hastings et al., 2020). Here, we advanced this model system and identified a non-genetic mechanism of resistance that drives recovery and regrowth in a subset of cells. Using RNAseq and a suite of biosensors to track single-cell fates both in vitro and in vivo, we identified that early S phase cells have a greater ability to maintain proliferative capacity, which correlated with reduced DNA damage over multiple generations. In contrast, cells in G1, late S or those treated with PARP/RAD51 inhibitors, maintained higher levels of DNA damage and underwent prolonged S/G2 phase arrest and senescence. Combined with our previous work, these data indicate that there is a non-genetic mechanism of resistance in human lung adenocarcinoma that is dependent on the cell cycle stage at the time of cisplatin exposure.

## Introduction

Lung adenocarcinoma (LUAD) is the most common form of lung cancer and the leading cause of cancer-related death in Australia. Less than 15% of patients have a targetable driver mutation and therefore cannot benefit from targeted therapy (*Herbst et al., 2018*). Consequently, the overwhelming majority of LUAD patients receive platinum-based chemotherapy as standard of care. The anti-tumour abilities of platinum compounds were first identified over 50 years ago with the discovery of cisplatin (*Kelland, 2007*). Since then, cisplatin and its derivatives have become one of the most successful groups of chemotherapeutics ever developed. Platinum therapy is essentially curative in testicular cancer, with survival rates > 90%, and is also a frontline treatment for small-cell lung cancer, ovarian, head and neck, bladder, and cervical cancers (*Gonzalez-Rajal et al., 2020*; *Kelland, 2007*).

Unfortunately, response rates to platinum in LUAD are below 30%, due primarily to innate resistance (*Herbst et al., 2018*). Nearly 150 different mechanisms of platinum resistance have been identified to date (*Stewart, 2007*). The vast majority of these mechanisms have been derived from preclinical models that utilise continuous, high-dose-exposure models, well above what is physiologically achievable in patients. Unsurprisingly, the majority of these models have failed to translate into improved clinical outcomes. To overcome this, we recently demonstrated that analysis of an in vitro assay that accurately models the in vivo drug exposure kinetics for cisplatin could provide therapeutically relevant insights into the signalling dynamics associated with innate resistance (*Hastings et al., 2020*). Cisplatin is given to patients as a single bolus dose, reaching a peak plasma concentration of ~14 μM (5 μg/ml), which is then rapidly cleared by the kidneys within 2–4 hr (*Andersson et al., 1996*; *Urien and Lokiec, 2004*). We therefore mimicked this in vitro by pulsing cells for 2 hr with the maximum plasma concentration (*Hastings et al., 2020*).

Once inside cells, platinum compounds can bind to DNA, RNA, and proteins (*Gonzalez-Rajal et al., 2020*); however, the binding to DNA, which forms platinum-DNA adducts, is thought to be the primary mechanism for their tumour-specific killing. Intra-strand DNA-platinum adducts are repaired by base excision and nucleotide excision repair during G1 (*Slyskova et al., 2018*). Inter-strand crosslinks (ICLs) are removed largely by the Fanconi anaemia (FA) pathway (*Smogorzewska, 2019*), which generates single- and double-strand breaks that are resolved by either the high-fidelity homologous recombination (HR) pathway during S phase (*Karanam et al., 2012*) or by the error-prone non-homologous end joining (NHEJ) pathway during G1 and G2 phase (*Enoiu et al., 2012*; *Slyskova et al., 2018*). Consequently, targeting DNA repair pathways has become a major focus for enhancing platinum chemotherapies. For example, cells with defective HR repair have been shown to be highly sensitive to combination therapy with cisplatin and PARP inhibitors in a number of cancer types, including ovarian and breast (*Tutt et al., 2018*). However, correlation between cisplatin sensitivity and impaired DNA repair has often failed to translate clinically in LUAD (*Mamdani and Jalal, 2016*). In contrast, we have recently identified TGF-β (*Marini et al., 2018*) and P70S6K (*Hastings et al., 2020*) as key mediators of innate platinum resistance in LUAD. We now build upon these previous results and identify in this research advance that a sub-population of cells are capable of continued proliferation despite exposure to pulsed cisplatin. Using a combination of cell cycle, DNA damage, and replication biosensors together with real-time single-cell fate tracking, we identified that these proliferative cells were enriched in late G1/early S phase at the time of cisplatin exposure and were able to sufficiently repair their DNA over multiple generations and rounds of replication. These results increase our understanding of the complexities underlying non-genetic resistance and recovery mechanisms in LUAD, while also highlighting mechanistic issues with a number of current clinical trials focused on combination therapy with cisplatin.

## Results

### Cells remain equally sensitive upon re-exposure to pulsed cisplatin

In our previous work (*Hastings et al., 2020*), we identified several targetable signalling pathways that were associated with resistance to cisplatin in LUAD cells. In this work, our goal was to analyse the innate mechanisms that enable cell survival after the initial exposure to cisplatin. To assess this, we analysed how cells respond to a subsequent dose of cisplatin, following recovery from an initial exposure event. To do this, we pulsed cells with cisplatin (5 μg/ml) and followed their response by time-lapse imaging. Cells were allowed to recover for 21–42 days (depending on their base rate of proliferation), before being challenged again with cisplatin, which equates to the approximate time patients normally receive a second dose in the clinic (*Figure 1A*). We utilised three LUAD cell lines: A549 (wild-type p53), NCI-H1573 (p53$^{R248L}$ mutant), and NCI-H1299 (p53 null), which were all engineered to stably express histone H2B fused to mCherry, allowing real-time quantitation of cell number and nuclear size. The initial pulse of cisplatin blocked the proliferation of A549 and NCI-H1573 cells, and significantly reduced NCI-H1299 cell numbers over a 3-day period. This was mirrored in colony formation assays, with both A549 and NCI-H1573 showing strong suppression of colony outgrowth, while p53-null H1299 cells were impacted to a lesser degree (*Figure 1—figure supplement*

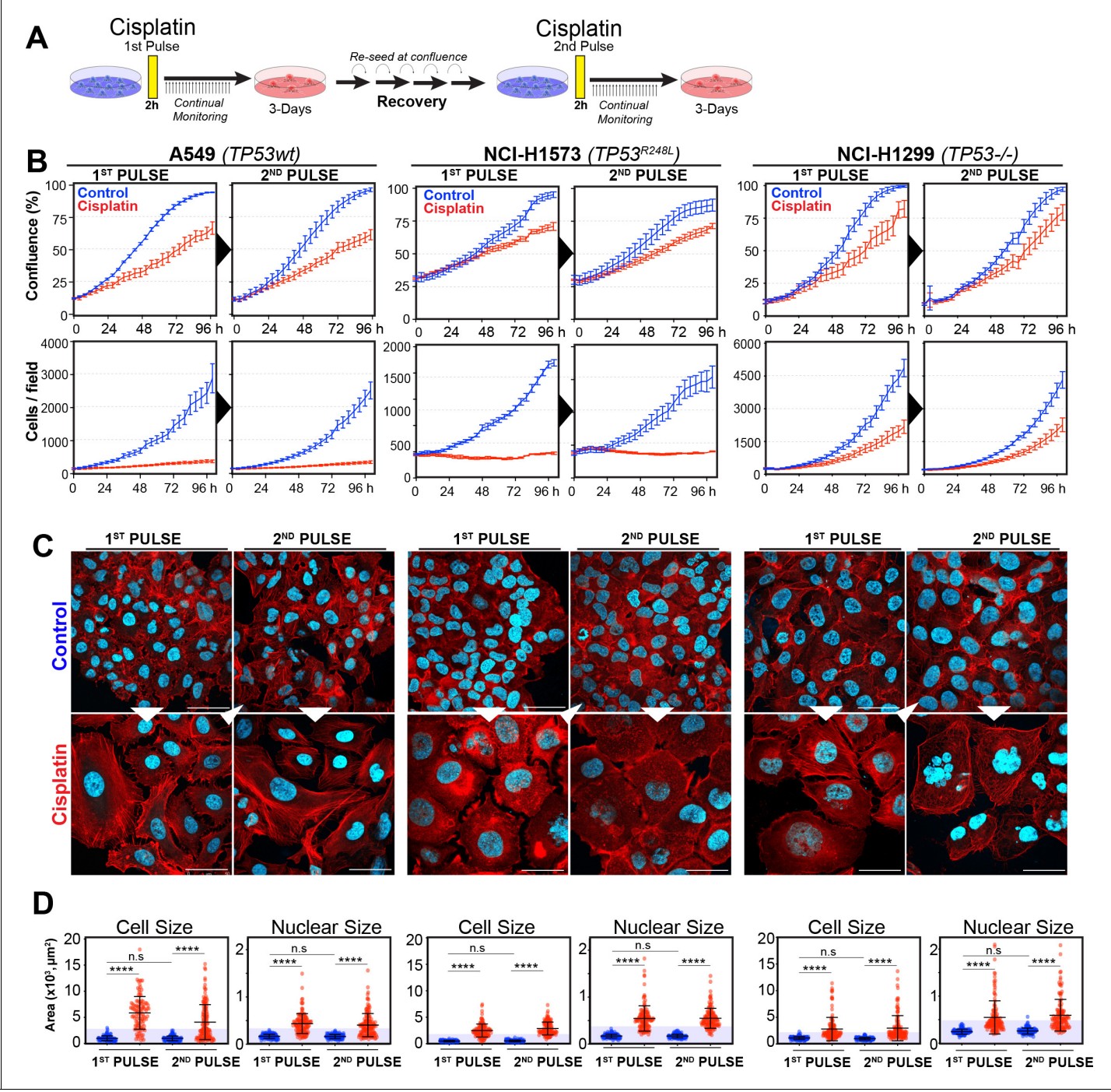

**Figure 1.** Rechallenging lung adenocarcinoma (LUAD) cells with cisplatin results in similar response profiles. (**A**) Schematic of rechallenging experiments. Briefly, cells stably expressing H2B-mCherry were pulsed with 5 µg/ml cisplatin for 2 hr. Cell proliferation (nuclear number and cell size) was then tracked for up to 4 days. Cells were then allowed to recover for 21 (A549 and NCI-H1299) or 42 days (NCI-H1573), re-culturing once confluent, before being re-pulsed with cisplatin. (**B**) Cell confluence and cell number were tracked for up to 4 days using IncuCyte based time-lapse imaging. Shown are the mean ± SD of n = 3 biological repeats. (**C**) Immunofluorescence of cells at 72 hr post-cisplatin treatment. Nuclei: cyan; Phalloidin-Alexa 647: red, scale bar = 10 µm. (**D**) Quantification of cell size and nuclear size from (**A**), with a minimum of n = 200 cells analysed per condition. Shown are the mean ± SD. Statistical significance was determined by one-way ANOVA (****p<0.0001, n.s = not significant).

The online version of this article includes the following figure supplement(s) for figure 1:

**Figure supplement 1.** Recovery of LUAD cells following cisplatin pulse.

*1A*). Interestingly, there was a less noticeable effect on cell confluence (*Figure 1B*). Subsequent visual and quantitative analysis of cells by immunofluorescence revealed a corresponding two- to six-fold increase in total cell and nuclear area across all three cell lines (*Figure 1C, D*), accounting for the reduced impact on confluence. Over the following 7–21 days of recovery (42 days for NCI-H1573 due to slower rate of proliferation), cells eventually recovered to their pre-pulse size and normal cell cycle distribution (*Figure 1C, D*, *Figure 1—figure supplement 1B*). In all three cell lines, subsequent cisplatin pulse of recovered cultures resulted in a near identical response to the initial pulse, both in terms of inhibition of cell number, reduced confluence, and increased cell size (*Figure 1B–D*). Based on these results, and our previous observation that all cells contained significantly increased levels of cisplatin-DNA adducts (*Hastings et al., 2020*), we concluded that cells surviving the first exposure remained equally sensitive to cisplatin and were therefore unlikely to have acquired resistance or arisen from an intrinsically resistant sub-clonal population within each cell line.

To assess this, we analysed the variability of cell and nuclear size after the initial pulse of cisplatin at 3–7 days post exposure. Visual analysis identified several colonies of cells whose size was similar to that of untreated control cells (*Figure 2A, B*). We hypothesised that these cells were able to pro-liferate and outgrow the non-proliferative (arrested) cells over the 21-to-42-day period. In support, a significant increase in senescence associated beta-galactosidase (β-gal) staining was observed both visually and by flow cytometry (C12FDG) in A549 and to a lesser extent H1573 and H1299 cell lines (*Figure 2—figure supplement 1A, B*). Increased levels of the cyclin-dependent kinase inhibitor p16, which is commonly linked with senescence, was associated with larger cells in NCI-H1573 and H1299 cells, while in A549 cells, which are null for p16 (*Kawabe et al., 2000*), p21 was similarly increased (*Figure 2—figure supplement 1C*). These data suggest that in each cell line the larger cells have a lower proliferative capacity and are likely senescent.

To test whether the surviving population arose from a subset of proliferating cells, we utilised the LeGO RGB colour-guided clonal cell tracking system (*Weber et al., 2011*). Briefly, each cell line was co-transfected with three different lentiviral vectors containing either a red, green, or blue fluores-cent protein. Each cell randomly received a variable amount of each plasmid resulting in a unique colour code for each cell. Quantitative colour analysis of untreated control cells revealed that up to 64 unique colours could be detected in A549 and NCI-H1299 cells and up to 46 colours in NCI-H1573 cells (*Figure 2—figure supplement 1D*). After pulsed exposure to cisplatin, single-colour col-ony outgrowths were clearly visible in all three cell lines at 3–7 days post exposure, which was main-tained at 21 for A549 and NCI-H1299 or 42 days for NCI-H1573 cells (*Figure 2C, D*). Colour diversity in cisplatin-recovered cells correlated with the colony formation assay data (*Figure 1—fig-ure supplement 1A*), with NCI-H1299 (p53 null) cells displaying a greater variability in recovered clonal colours compared to A549 (p53 wt) and NCI-H1573 (p53 mutant) cells (*Figure 2C, Figure 2—figure supplement 1D*). Taken together, these results indicate that repopulation of the culture after the initial pulse exposure is primarily driven by a small fraction of cells, with the total number of clones impacted by p53 status.

To confirm these results in vivo, we injected A549, NCI-H1573, or NCI-H1299 cells subcutaneously into the flanks of nude mice and allowed establishment of tumours (150 mm$^3$) before administering a single treatment of carboplatin (60 mg/kg). Mice harvested at 3 days post treatment were analysed by imunohistochemistry (IHC) for cell size and proliferating cell nuclear antigen (PCNA)-positive staining, with all cell lines showing a significant increase in cell size (*Figure 3A, B*). Similar to the in vitro results, active proliferation (PCNA positivity) in A549 cells was strongly suppressed after carbo-platin exposure, indicating that the majority of cells were not proliferating. In p53 mutant NCI-1573, there was no significant reduction in PCNA in vivo despite significant reduction in cell numbers in vitro cells. In contrast, NCI-H1299 cells, which lack p53, did not show any significant decrease in PCNA staining in vivo (*Figure 3A, B*). BrdU pulse labelling of in vitro-treated cells showed similar results, with strong suppression of active BrdU incorporation in A549 cells, with partial and no signifi-cant inhibition seen in NCI-H1573 and NCI-H1299 cells, respectively (*Figure 3—figure supplement 1*). In summary, these in vitro and in vivo data suggest that in LUAD cells exposure to pulsed cis-platin results in a significant proportion of cells increasing in size, with presence of functional p53 correlating with increased senescence and reduced active replication in enlarged cells. Interestingly, in all three lines, a sub-population of cells remain at a normal (stable) cell size, maintained their pro-liferative capacity, and drove repopulation of the cell culture. Despite this, these proliferative cells

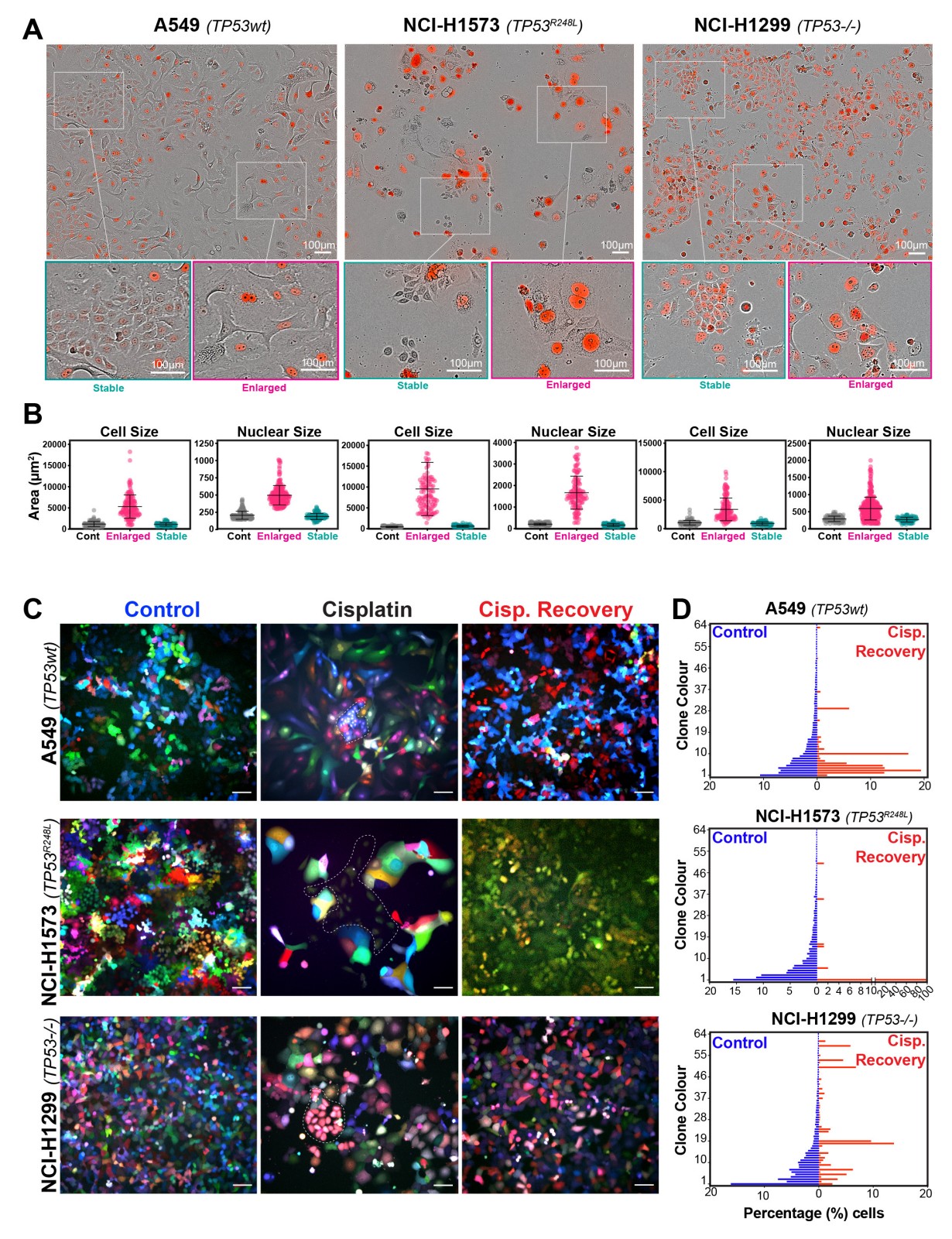

**Figure 2.** Variable cell size and clonal outgrowth in post-cisplatin-treated cells. (**A**) Representative images from cells treated as per *Figure 1A* showing control (Cont.), proliferative (Prolif.), and arrested (Arrest) cells. Scale bars = 100 μm. (**B**) Cell and nuclear size were calculated on sub-populations of cells that appeared to form clonal outgrowths. Quantification of cell size and nuclear size from (**A**), with a minimum of n = 100 cells analysed per condition. Shown are the mean ± SD. Statistical significance was determined by one-way ANOVA (****p<0.0001, n.s = not significant). (**C**) LeGO RGB

*Figure 2 continued on next page*

*Figure 2 continued*

colour-guided clonal cell tracking system was used to track clonal dynamics after cisplatin pulse treatment. Cells were treated as per *Figure 1A*, with clonal identification and quantification (D), measured at 3 and 21 days post-cisplatin exposure using Image J/Fiji (42 days post-cisplatin exposure for NCI-H1573).

The online version of this article includes the following figure supplement(s) for figure 2:

**Figure supplement 1.** Senescence and LeGo Clonal Recovery of LUAD cells following cisplatin pulse.

remained equally sensitive to subsequent cisplatin treatment, indicating a non-genetic mechanism of resistance.

## Differential RNAseq analysis of cisplatin-treated sub-populations

To better understand the potential mechanism driving the difference between cells that become enlarged and senescent compared to those that maintain stable size and proliferative capacity, we performed RNAseq analysis on each unique population. Briefly, A549 cells were pulsed with cisplatin, harvested at 72 hr, and sorted into stable or enlarged cells based on size. This was determined by forward scatter (FSC) and side scatter (SSC) parameters, with gates established based on the size of untreated control cells (*Figure 4—figure supplement 1A*). Pre- and post-sorted cells were then processed for RNAseq analysis (*Figure 4A, B*). Two-way hierarchical clustering indicated that there were clear differences in gene expression between cisplatin-treated cells that maintained a stable size compared to untreated control and enlarged cisplatin-treated cells (*Figure 4C*, *Supplementary files 1–4*). To better understand these effects, we undertook a more detailed bioinformatic analysis using Ingenuity Pathway Analysis (IPA). Strong upregulation of the CDK inhibitor p21 was present in both pre-sorted cisplatin-treated and post-sorted enlarged cells, matching the early flow data (*Figure 2—figure supplement 1C*). This corresponded with upregulation of *p53*, *CHK,* and G2/M cell cycle checkpoint signalling, and a corresponding reduction in DNA replication and increase in senescence pathways (*Figure 4D, E*), correlating with the increased β-gal and reduced proliferation observed above (*Figure 1B*, *Figure 2—figure supplement 1B*). Importantly, stable (size) cisplatin-treated cells were significantly different from untreated controls, indicating that these cells were impacted by cisplatin exposure, similar to our previous reports where all cells contained detectible cisplatin-DNA adducts post-pulsed exposure (*Hastings et al., 2020*). Notably, these stable cells displayed a decrease in *EIF2*, *mTOR,* and *p70S6K* signalling, with a corresponding increase in *TREM1*, *GP6,* and *IL-17F* signalling pathways (*Figure 4F*, *Supplementary files 3* and *4*). Four-way comparative analysis further highlighted key differences between each sub-population, with stable cells all showing strong upregulation of *BRCA1*, *ATM,* and DNA replication signalling pathways, which were heavily suppressed in non-proliferative cells (*Figure 4G*). Similarly, Gene Set Enrichment Analysis (GSEA) identified strong enrichment for cell cycle, HR-directed repair, *ATR,* and the Fanconi pathway in stable compared to enlarged cells (*Figure 4—figure supplement 1B, C*). In summary, these data suggest that the cisplatin-treated stable cells are considerably different from enlarged and untreated control cells. Specifically, stable cells do not undergo significant *p53/p21*-dependent cell cycle checkpoint arrest but do show prominent upregulation of DNA repair pathways involving *HR/BRCA1* and *ATM/ATR*.

## Cell cycle and p53 status at the time of exposure correlates with cell fate outcomes

The above RNAseq data indicated that there were strong cell cycle-dependent differences between proliferative and non-proliferative cisplatin-treated cells. To better understand these differences, we utilised the FUCCI biosensor system to enable real-time cell cycle status of individual cells, as published in the previous paper (*Hastings et al., 2020*). Briefly, asynchronous A549 cells stably expressing FUCCI were pulsed with or without cisplatin and then followed by time-lapse microscopy for 72 hr. Individual cells were manually tracked and scored for cell cycle status and cell fate as previously described (*Caldon and Burgess, 2019*; *Hastings et al., 2020*). The majority of control cells divided at least two times within the 72 hr time period (*Figure 5A*, *Figure 5—video 1*). In contrast and as demonstrated previously (*Hastings et al., 2020*), cisplatin-treated cells showed a range of cell cycle perturbations (*Figure 5B*, *Figure 5—figure supplement 1A*), including a significant and prolonged

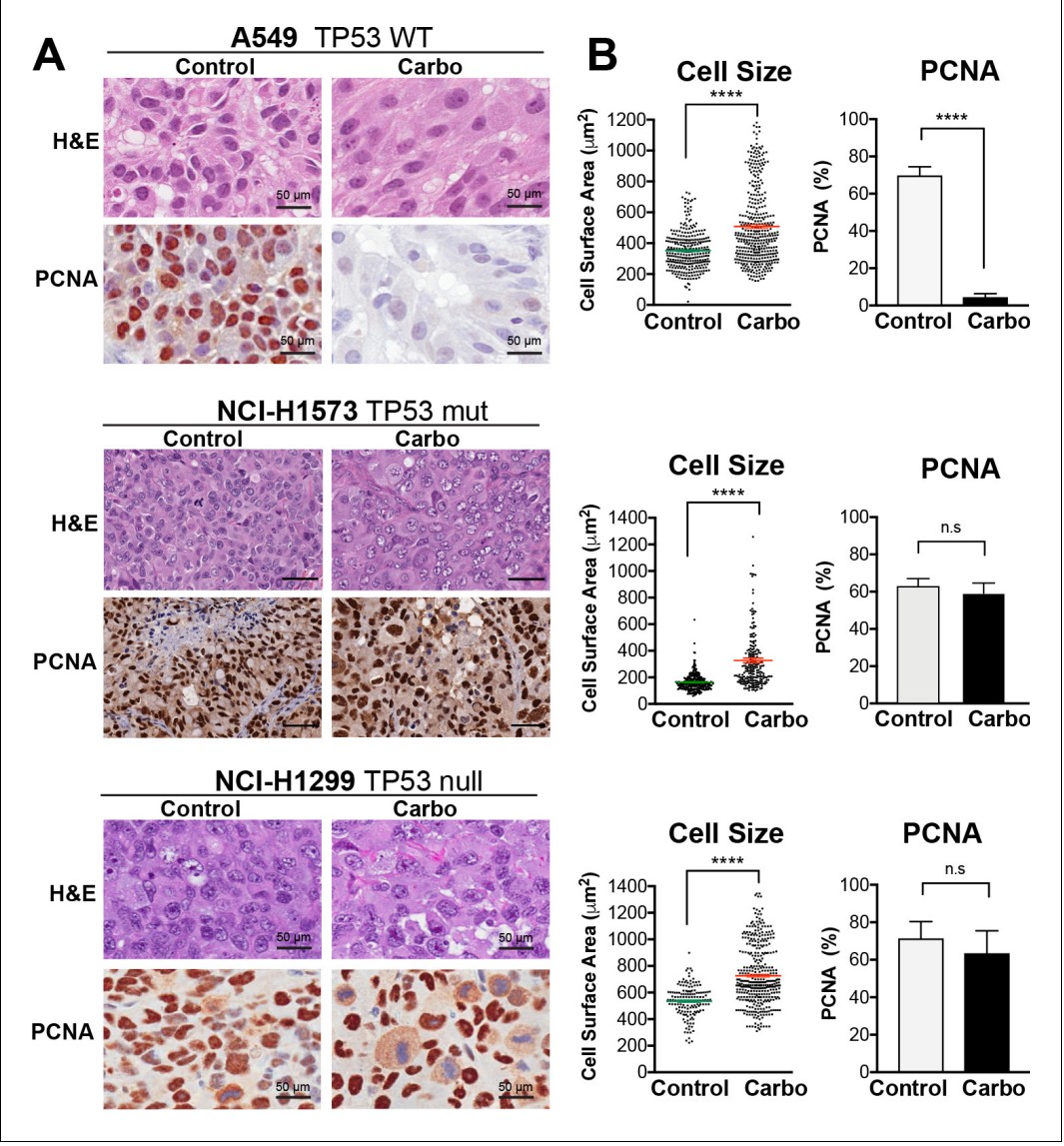

**Figure 3.** In vivo validation of cisplatin effects on cell size and proliferation. (**A**) A549 and NCI-H1299 cells were injected subcutaneously with $2 \times 10^6$ cells into the flanks of nude mice (n = 5). Carboplatin (60 mg/kg) was delivered by a single tail-vein injection, and tumours were harvested at 3 days post treatment and analysed by IHC for cell size and PCNA-positive staining. Scale bar = 50 μm. (**B**) Quantification of IHC images from (**A**) (control n = 300, carbo n = 400). Shown are the mean ± SD. Statistical significance was determined by unpaired two-tailed Students t-test (****p<0.0001, n.s = not significant).

The online version of this article includes the following figure supplement(s) for figure 3:

**Figure supplement 1.** Cells were pulsed cisplatin or not (0 hr) for 2 hr and then with BrdU for 2 hr prior to harvesting at the indicated timepoints.

S/G2 phase arrest (*Figure 5—figure supplement 1B*), which correlated with a reduced number of total divisions (*Figure 5—figure supplement 1C*). Combining this with additional scoring of cisplatin-treated cells that underwent multiple (two or more) divisions within the 72 hr period revealed an enrichment for cells that were in late G1 and early S phase at the time of cisplatin exposure (*Figure 5B*). Unbiased analysis of an additional 400 cisplatin-treated A549 cells found a significant increase in the number of divisions arising from cells that were in G1/S or early S phase at the time of exposure compared to G1 phase cells (*Figure 5C*). Furthermore, the overwhelming majority of cells in late S or G2/M only completed one division during the 72 hr period. Taken together, these

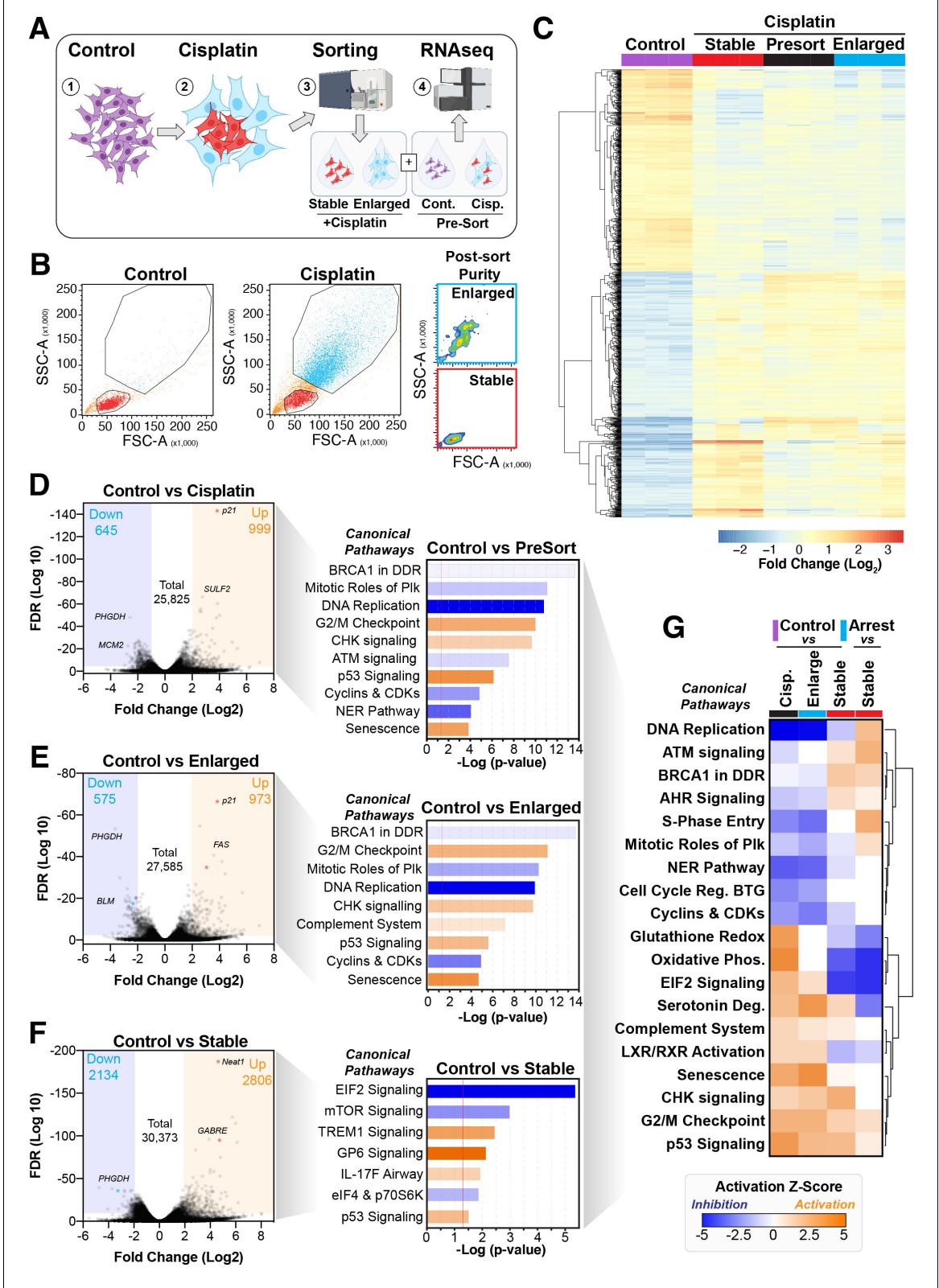

**Figure 4.** Comparative RNAseq analysis of FACS-sorted cisplatin-treated cells. (**A**) Schematic (created with BioRender.com) describing treatment, sorting, and analysis pipeline. (**B**) Representative example of pre- and post-sorted control and cisplatin-treated cells. (**C**) Hierarchical clustering of three independent biological repeat experiments of all altered genes identified by RNAseq. (**D–F**) Volcano plots displaying significantly downregulated (blue) or upregulated (orange) genes and subsequent Ingenuity Pathway Analysis (IPA) canonical pathway analysis. Predictions of inhibition (blue) or

*Figure 4 continued on next page*

*Figure 4 continued*

activation (orange) or no change (white) states are based on the IngenuityKnowledge Base, which compares the expected change with experimental observation to all known upstream canonical regulatory pathways. Variable stringent p-value (>1.3) and z-score (>0.5) cut-offs were used to limit pathways to top 7–8 most significant hits. (G) Hierarchical clustering of IPA comparative canonical pathway analysis.

The online version of this article includes the following figure supplement(s) for figure 4:

**Figure supplement 1.** FACS sorting and RNAseq analysis of A549 cells pulsed with cisplatin.

data suggest that there is a cell cycle-dependent mechanism driving the ability of A549 cells to continue to proliferate after cisplatin exposure.

Our above data indicated that loss of p53 increased the number of clones capable of regrowth after cisplatin exposure. To assess this in greater detail, we analysed asynchronous FUCCI expressing NCI-H1299 (p53 null) cells pulsed with or without (Control) cisplatin. Treatment with cisplatin significantly reduced the number of overall divisions in H1299 cells (*Figure 5—figure supplement 2A–C*); however, compared to A549 cells, this reduction was not as pronounced, in line with the cell proliferation data observed above (*Figure 1B*). Single-cell fate tracking of NCI-H1299 cells suggested a small bias for higher rates of division in cells that were in late G1 and early S phase at the time of treatment (*Figure 5—figure supplement 2B, C*); however, this trend was not significant (*Figure 5—figure supplement 2D*). We further assessed p53 loss by reanalysing our previously published data on siRNA knockdown of p53 in A549 cells (*Hastings et al., 2020*). Notably, depletion of p53 in A549 cells resulted in a corresponding increase in the number of divisions completed after pulsed exposure to cisplatin, supporting the NCI-H1299 data (*Figure 5—figure supplement 2E*).

To validate these results in vivo, we implanted FUCCI expressing A549, NCI-H1573, or NCI-H1299 cells under optical windows in mice (*Figure 5D*). Tumours were allowed to establish before mice were given a single dose of carboplatin. Individual mice were then repeatedly imaged over 7 days post treatment. Notably, prior to cisplatin treatment, approximately 70–80% of cells from each line were in G1 phase (*Figure 5E, F*, *Figure 5—figure supplement 3A, B*). Similar to in vitro results, we observed an increase in proportion of S/G2 phase cells at day 1 in all cell lines, indicating that cells were arrested in S/G2. In A549 cells, the percentage of S/G2 cells reduced gradually from day 3 to 7, resulting in over 90% of A549 cells in a G1 like state (red) at 7 days post treatment (*Figure 5E, F*). In contrast, the percentage of S/G2 cells in both NCI-H1573 or NCI-H1299 increased until day 3, before returning to pre-treatment levels by day 7 (*Figure 5—figure supplement 3A, B*). This mirrored our above in vitro data, where the presence of wild-type p53 (in A549 cells) corresponded with a higher rate of G2-exit and senescence compared to p53 null and mutant cells (*Hastings et al., 2020*), suggesting that the results we observed in vitro are recapitulated in vivo.

## Cisplatin treatment during late G1 early S phase correlates with multiple divisions

The above data indicated that in p53 wild-type cells those in late G1 and early S phase at the time of exposure had a greater capacity to undergo multiple division compared to cells G1 and S-G2/M phase. To assess this in greater detail, we synchronised and released FUCCI expressing A549 cells into either G1 or early S phase using either palbociclib or thymidine, respectively (*Figure 6A, B*, *Figure 6—video 1*, *Figure 6—video 2*). We combined these synchronisations with pulsed exposure to cisplatin at various points following release to target G1, early or late S phase populations, which were then monitored by time-lapse microscopy. To target G1, cells were pulsed with cisplatin upon release from palbociclib (Cis at G1). This resulted in the majority of cells undergoing a prolonged S/G2 phase and then exiting back into a G1-like state without undergoing mitosis (G2-exit; *Figure 6C*), a state we described previously (*Hastings et al., 2020*). Notably, only 7/50 cells completed a single division, and no cells underwent multiple (two or more) divisions during the 72 hr time period (*Figure 6A–D*; proliferative). In contrast, targeting cells in G1/S (Cis at G1/S) using either palbociclib or thymidine synchronisation with cisplatin resulted in significantly more cells (13/50 and 16/50, respectively) completing two or more divisions (*Figure 6A–D*; proliferative). Finally, nearly all cells targeted in late S phase (Cis at S) completed the first mitosis and then underwent prolonged S/G2 arrest and G2-exit, with only 4 out of 50 cells completing two divisions within the 72 hr period

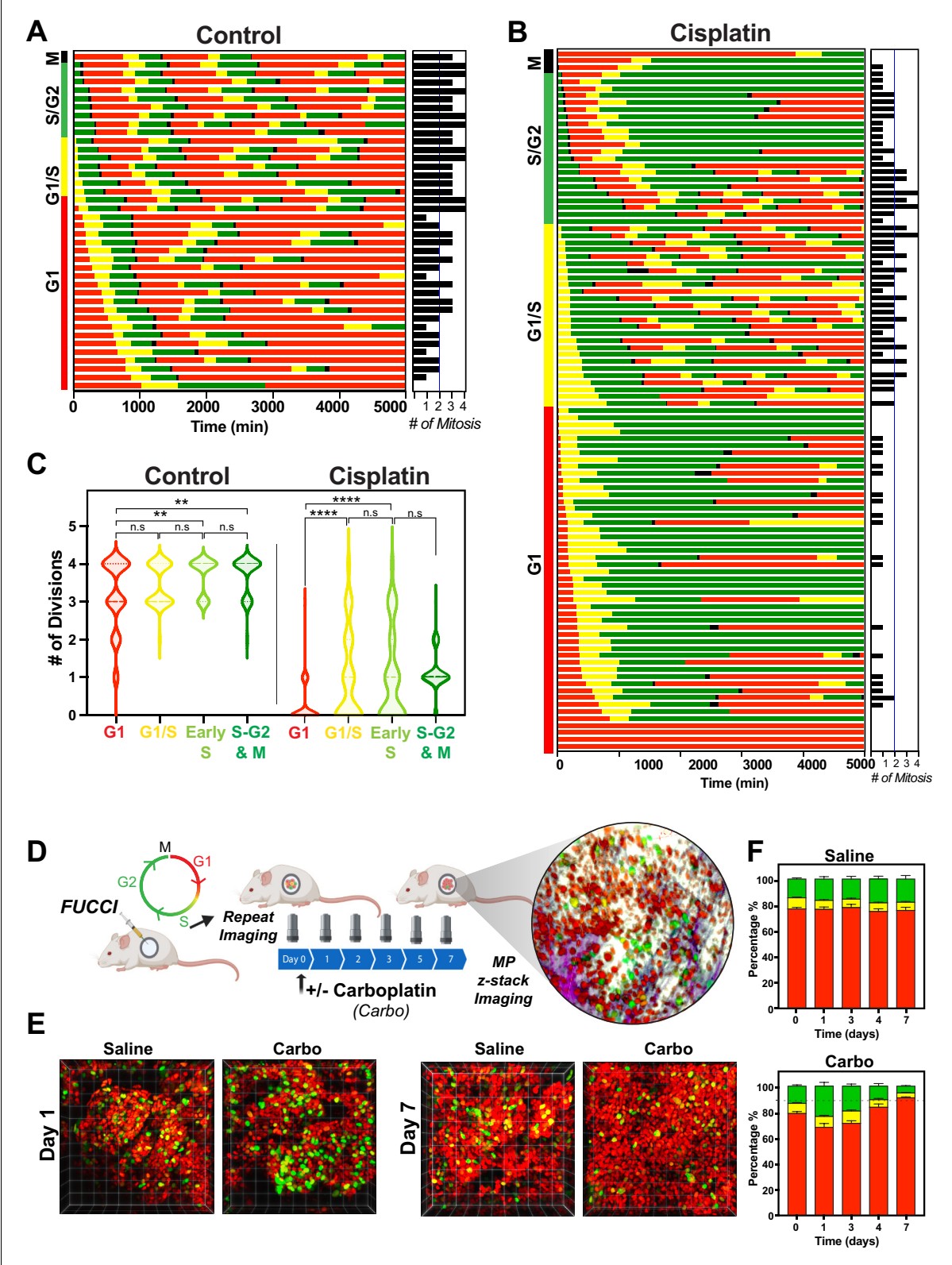

**Figure 5.** Cell cycle and p53 status at the time of exposure correlates with cell fate outcomes. (A, B) The fate of asynchronously growing FUCCI expressing A549 individual control (n = 50) and cisplatin pulsed cells (n = 100) was tracked by time-lapse microscopy, with images taken every 30 min for 72 hr. Cisplatin-treated cell analysis was slightly biased for cells that underwent multiple divisions. (C) Violin plots of the number of divisions from cells in (A, B), based on the cell cycle phase at the time of cisplatin exposure. Statistical significance was determined by one-way ANOVA with Tukey's

*Figure 5 continued on next page*

*Figure 5 continued*

correction for multiple comparisons (**p<0.01, ****p<0.0001, ns = not significant). (**D**) Schematic (created with BioRender.com) of optical window-based longitudinal in vivo imaging of FUCCI A549 cells. (**E**) Representative 3D projection images from mice imaged at days 1 and 7 with carboplatin (Carbo) or control (Saline). (**F**) Quantification of the proportion of red (G1), yellow (G1/S), and green (S/G2-M) cells found in tumours (n = 3) from day 0 to 7.

The online version of this article includes the following video and figure supplement(s) for figure 5:

**Figure supplement 1.** Additional replicates of FUCCI expressing A549 cells pulsed with cisplatin.
**Figure supplement 2.** Role of p53 in regulating response to pulsed cisplatin.
**Figure supplement 3.** In vivo analysis of FUCCI expressing LUAD cells.
**Figure 5—video 1.** Asynchronous A549 cells pulsed with or without cisplatin.
https://elifesciences.org/articles/65234#fig5video1

(*Figure 6B–D*). Taken together, these data indicate that cells in late G1 and early S at the time of cisplatin exposure have a greater capacity to continue proliferating.

## Disruption of DNA repair reduces ability of early S phase cells to proliferate

A major target of cisplatin is DNA, with intra-strand crosslinks and ICLs disrupting replication and repair, leading to stalled replication forks and the formation of double-strand breaks (*Gonzalez-Rajal et al., 2020*). We therefore hypothesised that cells in late G1/early S phase were able to repair cisplatin-induced DNA damage during the first cell cycle more efficiently than cells in early G1 or late S phase, thereby allowing them to continue proliferation. To test this, we engineered A549 cells to stably co-express a truncated form of 53BP1 fused to mApple (Apple-53BP1trunc), which has previously been shown to bind double-strand break sites co-marked with γH2A.X but lacks any of the functional domains of 53BP1 (*Yang et al., 2015*). We combined this with a PCNA chromobody, where we replaced GFP with mNeonGreen, to mark sites of active DNA replication (*Figure 7A*, inset). Cells were synchronised with thymidine and pulsed with cisplatin 2 hr prior to release to enrich for early S phase-targeted cells (as per *Figure 6B*), and then tracked by 4D live*cell imaging. Individual cells were divided into either those in G1, early S, or mid/late S based on the pattern of PCNA foci (cyan), and then tracked through time (*Burgess et al., 2012*; *Charrasse et al., 2017*). In control cells, a small number (<20) of 53BP1-positive foci (red hot) were observed as cells underwent the first round of replication (*Figure 7A, B*, *Figure 7—video 1*). Daughter and grand-daughter cells then displayed several (<5) large foci during G1 (up to 5 $\mu m^2$), which were resolved as cells entered S phase and began replicating (*Figure 7C*, white arrow). Cells that were in G1 at the time of cisplatin exposure entered S phase and rapidly accumulated a large number (~100) of 53BP1-positive foci; these slowly reduced over the remainder of the time lapse (*Figure 7A–C*), which corresponded with an increase in the average size of the foci (~1 $\mu m^2$). In contrast, cells that were in early S phase and completed multiple (two or more) divisions within the 72 hr timeframe showed a rapid rise in foci number (~100), which then decreased at the conclusion of S phase, correlating with an increase in foci size. A small number of larger foci were present in the following G1 cell, although the size of these foci was smaller than those observed in control daughter cells (~1 $\mu m^2$). Interestingly, in grand-daughter and great grand-daughter cells, the size of G1 foci increased (>2 $\mu m^2$), in line with G1 foci observed in control cells (*Figure 7A–C*, *Figure 7—figure supplement 1A*). Finally, cells in mid-late S phase also showed a large number of 53BP1 foci, which increased in size as cells progressed through the first G2 phase. Interestingly, the average number of 53BP1 foci in mid-S were higher and were removed later, just prior to mitotic entry compared to cells from early S phase (*Figure 7A–C*, *Figure 7—figure supplement 1B*). Furthermore, the quality of mitosis was often reduced in cells from mid-S phase, with cells displaying chromatin bridges, micronuclei, and/or failed cytokinesis (*Figure 7—figure supplement 1C*), correlating with the increase in death during or after mitosis we observed previously (*Figure 6C*). The subsequent daughter cells from those exposed in mid-S phase then showed a rapid rise in the number of foci (>100) as they began replication. In contrast, the number of 53BP1 foci in early S phase cells only increased mildly during replication and was notably lower than the numbers observed in the first round of replication (*Figure 7—figure supplement 1A, B*). Based on these results, we concluded that early S phase cells were able to either partially repair double-strand breaks during the first round of DNA replication and/or mark damage for

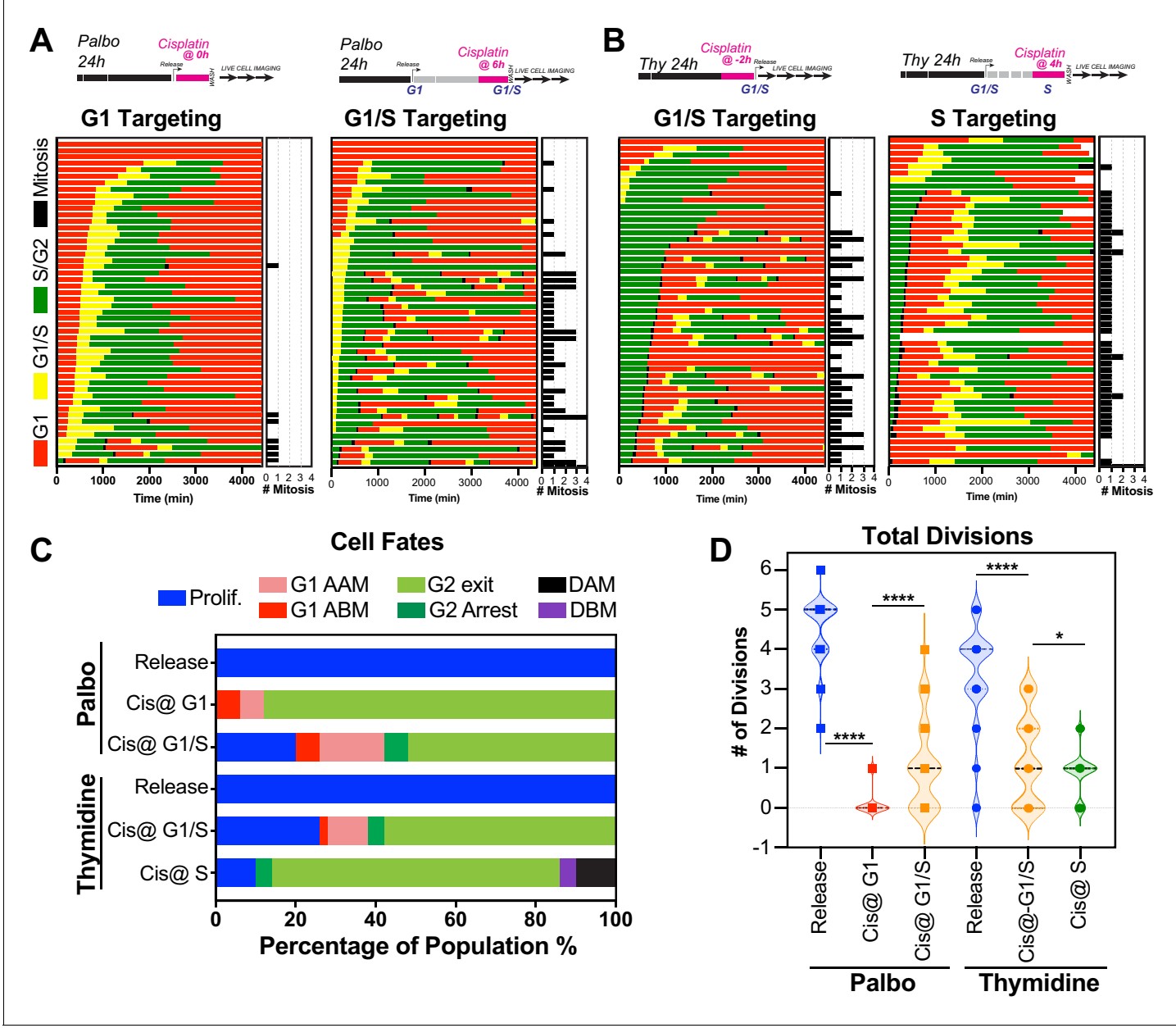

**Figure 6.** Cell cycle-dependent regulation of cisplatin response. (**A, B**) Schematic of palbociclib (Palbo) and thymidine (Thy) protocols used to synchronise FUCCI expressing A549 cells in G1, G1/S, and S phase prior to 2 hr cisplatin pulse treatment. Specifically, G1 cells were released from palbociclib and pulsed immediately with cisplatin (Palbo + Cis at G1). G1/S cells were pulsed with cisplatin at 6 hr post release from palbociclib (Palbo + Cis at G1/S), or alternatively G1/S phase cells targeted by treatment with cisplatin 2 hr prior to release from thymidine (Thy + Cis at G1/S). Finally, S phase cells targeted by pulsing with cisplatin at 4 hr post-thymidine release (Thy + Cis at S). The fate of individual cells (n = 50) was tracked by time-lapse microscopy, with images taken every 30 min for 72 hr. (**C**) Quantification of cell fate outcomes from (**A**), including G1 arrest before mitosis (G1 ABM), G1 arrest after mitosis (G1 AAM), death before mitosis (DBM), and death after mitosis (DAM) and proliferative (Prolif.). (**D**) Quantification of the total number of cell divisions observed in each condition (n = 50). Mean is shown, statistical significance was determined by one-way ANOVA with Tukey's correction for multiple comparisons (****$p<0.0001$, *$p<0.05$).

The online version of this article includes the following video(s) for figure 6:

**Figure 6—video 1.** Palbociclib sychronised FUCCI expressing A549 cells pulsed with cisplatin.
https://elifesciences.org/articles/65234#fig6video1

**Figure 6—video 2.** Thymidine sychronised FUCCI expressing A549 cells pulsed with cisplatin.
https://elifesciences.org/articles/65234#fig6video2

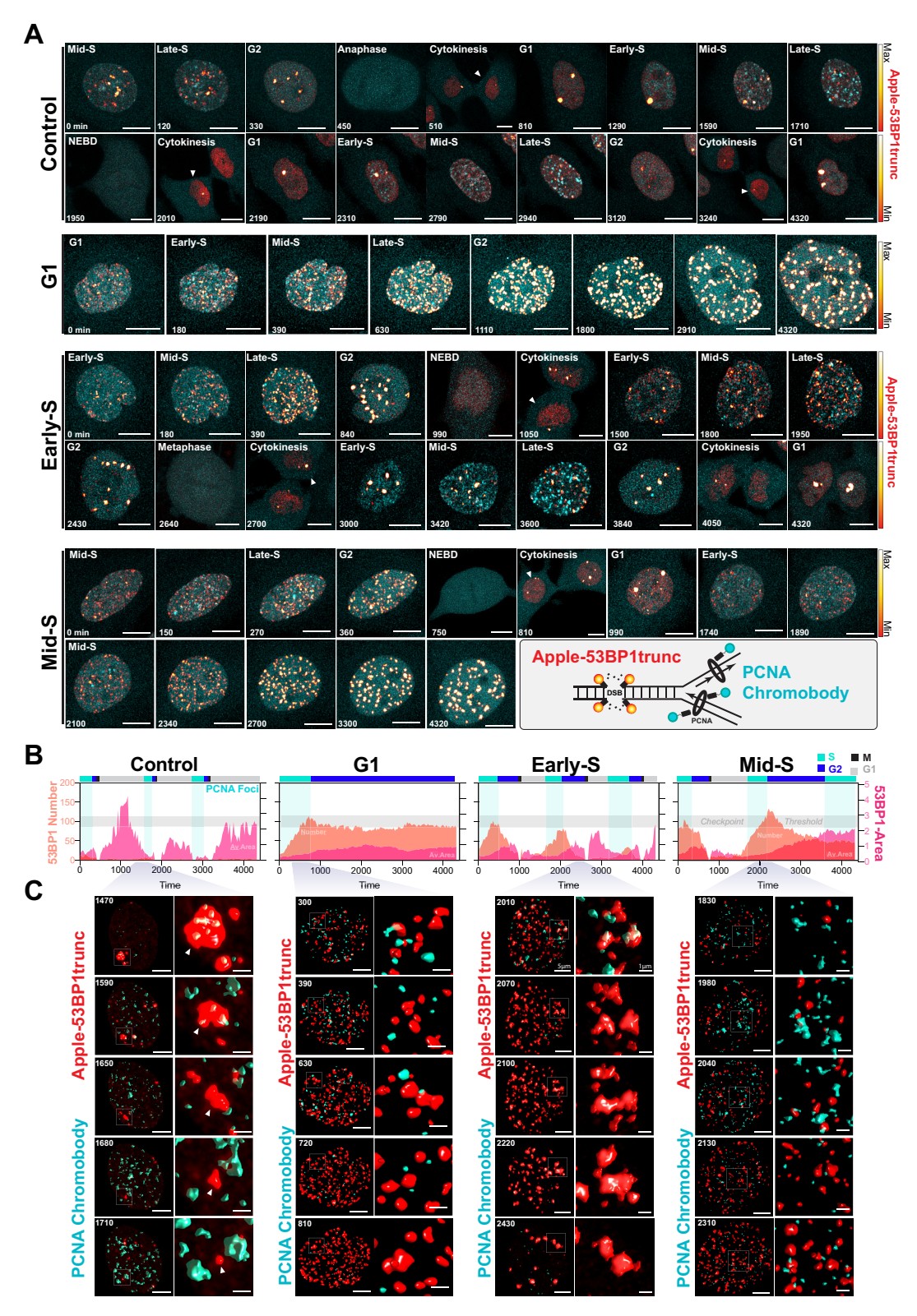

**Figure 7.** Dual DNA replication and damage biosensor analysis of cisplatin-treated cells. (**A**) Representative maximum image projections of A549 cells co-expressing a mNeonGreen tagged PCNA chromobody (cyan) and a truncated version of 53BP1 tagged with mApple (trunc53BP1-mApple; red-hot LUT). Cells were imaged using confocal microscopy, with 10-μm-thick z-stack (1 μm slice) taken every 30 min for 72 hr. Scale bar 10 μm. (**B**) Quantification of the size and number of PCNA and 53BP1 foci for each cell shown in (**A**). (**C**) 3D volume renders from cells in (**A**) for the indicated

*Figure 7 continued on next page*

*Figure 7 continued*

times, with cropped zoom areas (right image). White arrows indicate 53BP1 foci that reduce in size over time (min). Scale bars 5 µm and 1 µm for left and right panels, respectively.

The online version of this article includes the following video and figure supplement(s) for figure 7:

**Figure supplement 1.** Additional analysis of dual DNA replication and damage biosensor analysis of cisplatin-treated cells.

**Figure 7—video 1.** Dual DNA replication and damage biosensor analysis of cisplatin-treated cells.

https://elifesciences.org/articles/65234#fig7video1

efficient repair in the subsequent daughter and grand-daughter cells. In contrast, cells in G1 had much greater levels of damage and remained arrested in the first G2 phase. Cells in mid/late S phase completed the first division while acquiring damage but were unable to sufficiently repair the damage before mitosis. Consequently, daughter cells with unrepaired damage had increased rates of mitotic induced breaks as they attempted the second round of DNA replication, leading to a strong S/G2 phase checkpoint arrest, similar to cells initially exposed during G1.

Based on these results, we hypothesised that cells in early S phase were better able to take advantage of the high-fidelity HR pathway, whose activity peaks in mid-S phase (*Karanam et al., 2012*) compared to G1 or late S phase. To test this, we utilised the PARP inhibitor, olaparib, to trap PARP at single-strand break sites, leading to increased rates of replication fork stalling and reduced capacity to repair double strand breaks (DSBs) by HR (*Murai and Pommier, 2018*). We hypothesised that this would increase the rate of damage in all cells and reduce the ability of early S phase cells to repair during the first cell cycle. To test this, A549 FUCCI cells were targeted in G1/S phase with cisplatin by synchronising with palbociclib or thymidine, as previously described (*Figure 6A, B*). Cells were then treated with or without olaparib (PARPi) for 1 hr prior to pulsed cisplatin exposure and monitored by time-lapse microscopy (*Figure 8A*). In cells treated with cisplatin, co-treatment with PARPi significantly reduced the total number of divisions (*Figure 8B*), indicating that cells were unable to continue proliferating. This correlated with a trend toward G1 delay in palbociclib and significant G1 delay in thymidine-synchronised cells (*Figure 8C*). Interestingly, although co-treatment with PARPi decreased the percentage of proliferative (two or more divisions) cells, there was only a small increase in death observed (*Figure 8D*), indicating that PARPi alone is not sufficient to drive increased toxicity to cisplatin in A549 cells. Importantly, inhibition of PARP did increase the rate of 53BP1 foci formation compared to cisplatin alone in asynchronous cells (*Figure 8E*). Furthermore, this correlated with a significant increase in both the amount of γ-H2AX staining and the size of cells (*Figure 8F*). We further validated these results by inhibiting RAD51 with RI-1, to target HR-mediated repair. RAD51 inhibition (RAD51i) had no significant effect on cell growth compared to untreated controls (*Figure 8—figure supplement 1A*). Single-cell fate tracking of FUCCI expressing A549 cells revealed a small but significant decrease in the number of divisions completed by early S phase cells treated with RI-1 and cisplatin compared to cisplatin alone. This correlated with a reduction in percentage of cells completing two or more divisions (proliferative) and an increase in the percentage of cells undergoing a G2-exit phenotype (*Figure 8—figure supplement 1B–D*). Taken together, these data suggest that targeting DNA repair pathways during the first replication cycle results in more DSBs and pronounced S/G2 cell cycle checkpoint arrest leading to G2-exit, likely due to a reduced ability to repair DNA damage. This in turn reduces the ability of cells to undergo further rounds of replication and division.

## Discussion

In this work, we have identified a novel, non-genetic mechanism of resistance to platinum chemotherapy, which facilitates continued proliferation in a subset of LUAD cells after pulsed exposure to cisplatin. These cells eventually outgrow the majority of arrested cells over the course of 3 weeks in vitro. However, upon re-exposure, they remained equally sensitive, indicating that the mechanism of resistance is not hard-wired, nor did cells acquire resistance after the first exposure. Quantitative single-cell fate tracking revealed that cells in late G1/early S phase at the time of exposure had a greater proliferative capacity after pulsed cisplatin exposure. This suggests that cell cycle stage at the time of exposure impacts how cells respond to cisplatin. Interestingly, cisplatin treatment of

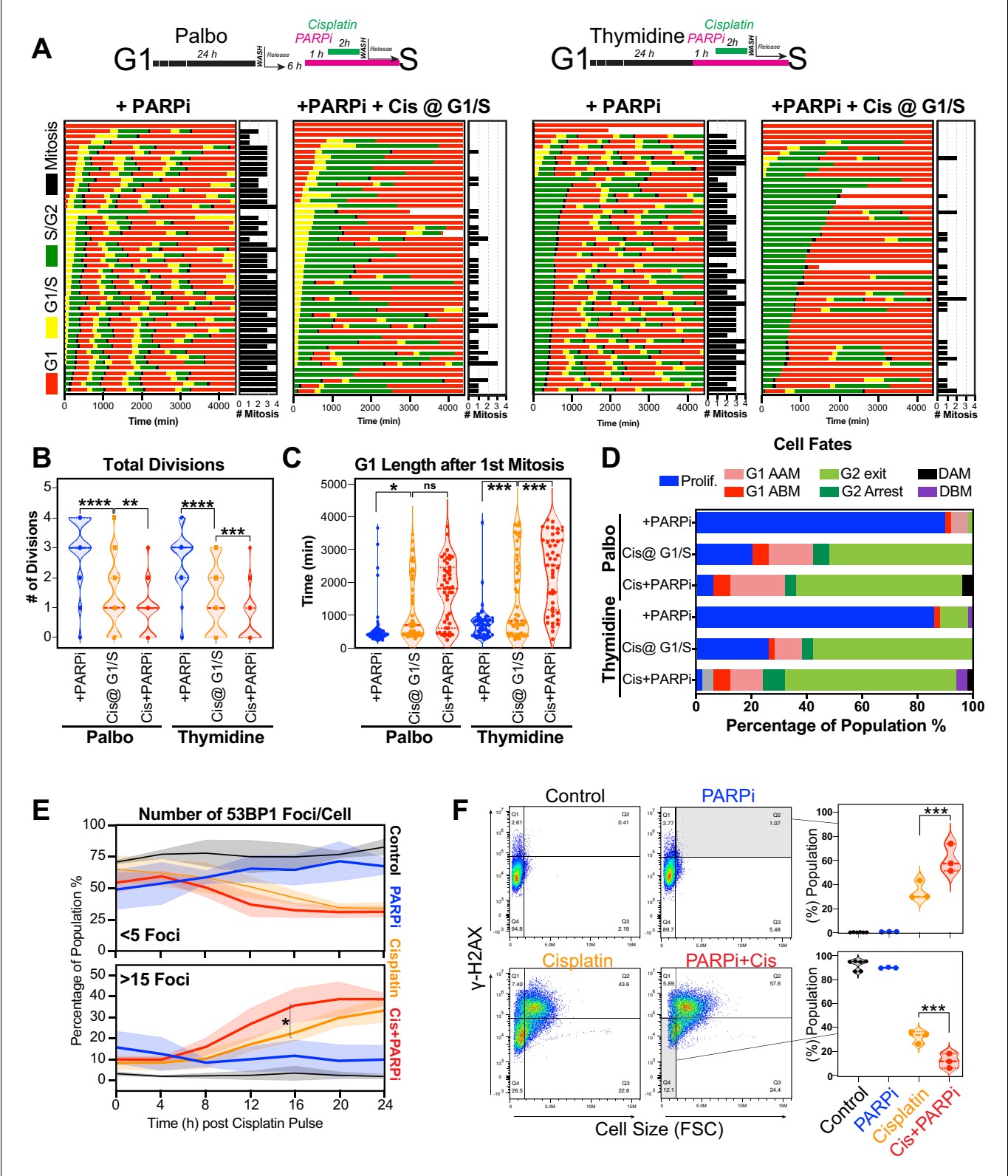

**Figure 8.** Inhibition of PARP reduces ability of early S phase cells to maintain proliferative capacity. (**A**) Schematic of palbociclib (Palbo) and thymidine (Thy) protocols used to synchronise FUCCI expressing A549 cells in G1, early and late S phase prior to olaparib (PARPi; 1 μM) and 2 hr cisplatin (5 μg/ml) pulse treatment. The fate of (n = 50) individual cells was tracked by time-lapse microscopy, with images taken every 30 min for 72 hr. Quantification of the total number of cell divisions (**B**) and G1 length after (**C**) first mitotic division observed in each condition. Statistical significance was determined

*Figure 8 continued on next page*

*Figure 8 continued*

by a one-way ANOVA test with correction for multiple comparisons (****p<0.0001, ***p<0.001, **p<0.01, *p<0.05). (D) Quantification of cell fate outcomes from (A), including G1 arrest before mitosis (G1 ABM), G1 arrest after mitosis (G1 AAM), death before mitosis (DBM), and death after mitosis (DAM) and proliferative (Prolif.), that is, cells that divided two or more times. (E) Fluorescent imaging of asynchronous A549 dual biosensor cells pulsed with cisplatin for 2 hr. The percentage of cells with <5 or >15 53BP1 foci/cell after cisplatin treatment are shown. A minimum of 250 cells per timepoint and condition were counted from (n = 3) biological repeats. Statistical significance was determined by two-way ANOVA (*p<0.05). (F) Thymidine-synchronised cells treated as per (A) were harvested and analysed for cell size and γ-H2AX by flow cytometry. Representative FACS plots and quantification from (n = 3) biological repeats are shown. Statistical significance was determined by one-way ANOVA with Tukey's correction for multiple comparisons (***p<0.001).

The online version of this article includes the following figure supplement(s) for figure 8:

**Figure supplement 1.** Inhibition of RAD51 reduces ability of early S phase cells to maintain proliferative capacity.

head and neck (UM-SCC-38) cancer cells resulted in similar heterogenous cell cycle and cell fate responses (*Luong et al., 2016*), implying that the non-genetic cell cycle-dependent mechanisms of resistance we observed here may translate to multiple cancer types beyond LUAD.

For cells that are in G1, which under normal physiological conditions represents the vast majority of cells (>70%) both in vitro and in vivo, the predominate response is to arrest in S/G2 during the first replication cycle (*Figure 9*). Notably, intracellular pH is lowest during G1, and cisplatin DNA binding is markedly increased in acidic conditions (*Stewart, 2007*), hence cells in G1 phase at the

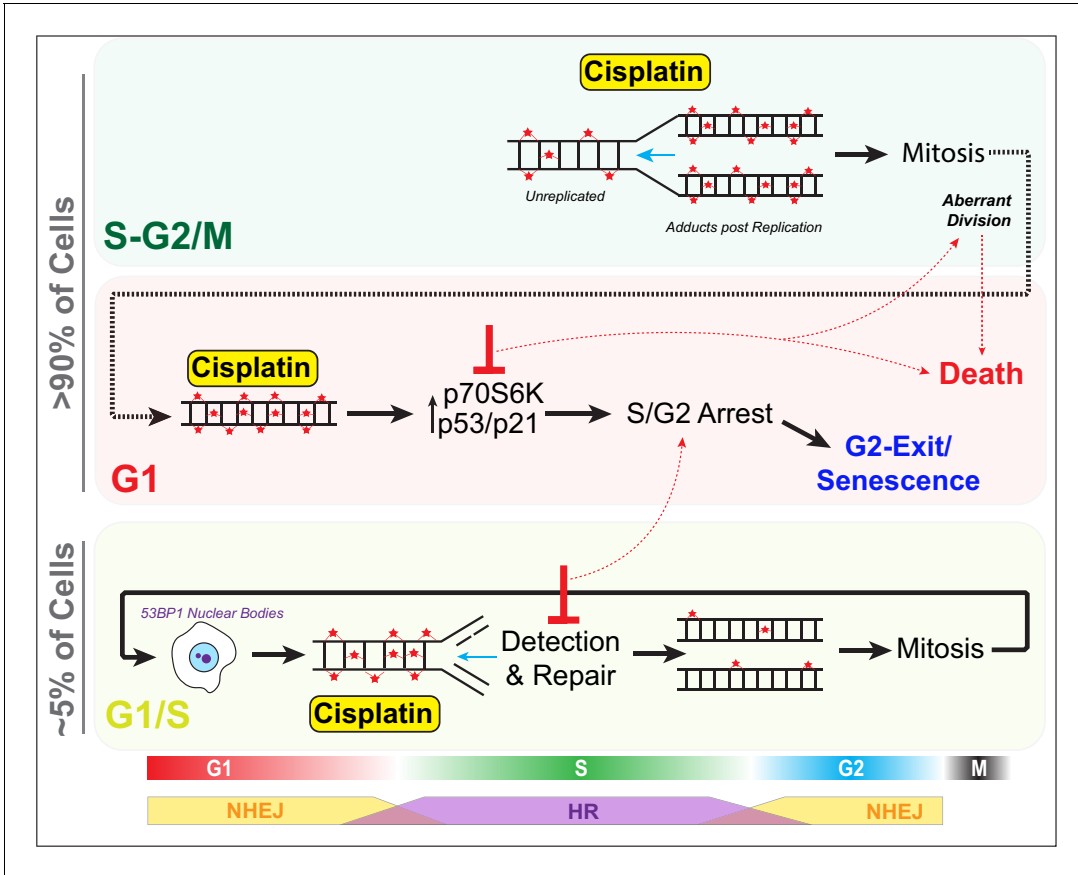

**Figure 9.** Schematic outlining cell cycle dependence of cisplatin recovery. Briefly, cells exposed in G1 undergo S/G2 phase arrest, marked by high levels of P70S6K, p53, and p21, which results in a stable and permanent cell cycle exit from G2 phase. Cells in late S phase likely receive platinum adducts in areas of already duplicated DNA, with error-prone non-homologous end joining (NHEJ) favoured over homologous recombination (HR). Combining this with an increase in cells undergoing aberrant division results in daughter cells displaying higher rates of damage during replication, further resulting in S/G2 arrest and senescence. In contrast, cells in late G1/early S phase have the opportunity to detect and repair damage by HR during the first cycle, thereby increasing chances that daughter cells will successfully complete and repair damage, allowing continued proliferation.

time of exposure may have higher levels of cisplatin-DNA adducts. In support, as G1-treated cells entered S phase, we noted increased numbers and size of 53BP1 foci, indicating widespread DSBs. This in turn correlated with prolonged S/G2 delays, indicating activation of DNA damage checkpoint signalling. Furthermore, we previously observed strong phosphorylation and activation of the G2 checkpoint proteins Chk1/2 across all LUAD cell lines after pulsed cisplatin exposure (*Hastings et al., 2020*). Similarly, RNAseq data from enlarged A549 cells showed consistent activation of G2/M checkpoint pathways. Notably, sustained S/G2 DNA damage has been shown to lead to ATR-mediated checkpoint signalling and sustained p21 expression, blocking CDK activity and preventing FOXM1-dependent G2 transcription required for mitotic entry (*Saldivar et al., 2018*). Prolonged p21 expression can lead to S/G2 checkpoint-arrested cells undergoing a G2-exit and senescence-like state (*Baus et al., 2003*; *Feringa et al., 2018*). This provides an explanation for why cells with a functional p53/p21 pathway were more likely to undergo G2-exit, had higher rates of senescence, and reduced numbers of cell divisions compared to those lacking p53.

In contrast, cells in mid-late S or G2/M phase at the time of exposure proceeded through the first division with minimal delays, although numerous chromatin bridges were observed between separating daughter cells, which correlated with increased rates of mitotic and post-mitotic death (*Figure 9*). Similarly, loss, mutation, or, as we demonstrated previously, suppression of p53 signalling through inhibition of P70S6K (*Hastings et al., 2020*) resulted in more cells entering mitosis and undergoing mitotic and post-mitotic death. This likely explains why cells that lacked p53 were more likely to undergo multiple aberrant divisions in vitro and maintain positive PCNA staining in vivo as they lacked the ability to instigate a stable S/G2 cell cycle checkpoint arrest. Notably, chromatin bridges are often indicative of decatenation failure and increased replication stress (*Sarlós et al., 2017*), and are commonly formed after DNA damage caused by chemotherapies such as cisplatin. These bridges are often repaired by HR during S phase (*Chan et al., 2018*; *Chan and West, 2018*), but if repair is not completed before mitosis, they can lead to increased rates of cytokinesis failure, chromosome instability, and cell death during or post-mitotic division (*Bakhoum et al., 2014*; *Burgess et al., 2014*; *Hayashi and Karlseder, 2013*). Importantly, chromosome instability has been linked to cancer heterogeneity, metastasis, and acquisition of chemoresistance (*Sansregret et al., 2018*), with single chromosome gains sufficient to reduce cisplatin-mediated cell death by delaying cells in G1 and slowing proliferation rate (*Replogle et al., 2020*), highlighting the complexity and inter-dependence between innate and acquired resistance mechanisms.

Thirdly, cells in late G1 and early S phase showed a greater propensity to undergo multiple divisions following exposure to cisplatin. This correlated with lower levels of 53BP1 foci during the first round of replication, which reduced further in subsequent daughter and grand-daughter cells. A likely explanation is that early S phase cells would encounter cisplatin adducts during the first replication cycle, initiate repair using high-fidelity HR (*Figure 9*), which peaks during mid-S phase (*Karanam et al., 2012*). In contrast, error-prone NHEJ, which is preferred during G1 and G2, is likely favoured by cells exposed during G1 and late S phase (*Karanam et al., 2012*). In support, RNAseq analysis showed enrichment for HR-associated BRCA1-mediated signalling pathways in stable-sized cells compared to enlarged cisplatin-treated cells. It would be of interest to confirm this through future analysis of transcriptomes of cell cycle phase sorted cells. Consequently, cells exposed during early S phase likely have more time and capacity to detect and accurately repair damage, thereby avoiding chromatin bridges during mitosis compared to those treated later in S phase. In support, the 53BP1 foci resolved more quickly prior to the first mitosis in early S phase-treated cells. Post-mitotic G1 phase 53BP1 nuclear bodies increased in size from daughter to grand-daughter cells, indicating increased efficiency in identification and corralling of unrepaired DNA damage during the previous replication cycle. These G1 53BP1 nuclear bodies prevent daughter cells from encountering damaged DNA during replication (*Watts et al., 2020*), thereby avoiding deleterious fork stalling. For under-replicated DNA, this provides the cell with a second chance at repair (*Spies et al., 2019*). The partial repair in the first cell cycle also likely ensures a pulsed p21 response (*Hsu et al., 2019*) with lower levels preventing arrest in the following G1 phase (*Barr et al., 2017*), helping promote continued proliferative capacity.

Finally, this work also indicates the potential for complications when co-administering platinum chemotherapies with other targeted and cell cycle altering therapies such as palbociclib. Specifically, pre- or co-administration of palbociclib with platinum may inadvertently synchronise cells in late G1/early S phase, resulting in a higher proportion of cells repairing the damage during the first cell cycle

and maintaining proliferative capacity. In support, current clinical trials in head and neck cancer where both agents were given at the same time have resulted in no improvement to cisplatin therapy and resulted in significant treatment-related toxicity (*Swiecicki et al., 2020*). Conversely, administering palbociclib after cisplatin (and other chemotherapies) improves response and notably represses HR-dependent DNA repair (*Salvador-Barbero et al., 2020*).

In summary, this work increases the understanding of the mechanisms driving recovery from cisplatin treatment and identifies the need for novel combination therapies that not only enhance cell death, but also prevent non-genetic, cell cycle-dependent resistance mechanisms.

# Materials and methods

## Key resources table

| Reagent type (species) or resource | Designation | Source or reference | Identifiers | Additional information |
|---|---|---|---|---|
| Cell line (*Homo sapiens*) | A549 | ATCC | CCL-185, RRID:CVCL_0023 | |
| Cell line (*Homo sapiens*) | NCI-H1299 | ATCC | CRL-5803, RRID:CVCL_0060 | |
| Cell line (*Homo sapiens*) | NCI-H1573 | ATCC | CRL-5877, RRID:CVCL_1478 | |
| Transfected construct (human) | mVenus-hGeminin (1/110) (plasmid) | *Sakaue-Sawano et al., 2008* | | |
| Transfected construct (human) | mCherry-hCdt1 (30/120) (plasmid) | *Sakaue-Sawano et al., 2008* | | |
| Transfected construct (human) | Cell Cycle-Chromobody plasmid | Chromotek | ccr | Recloned into a pLVX lentiviral backbone with TagRFP replaced with mNeonGreen |
| Transfected construct (human) | 53BP1trunc-Apple | Addgene | 69531, RRID:Addgene_69531 | 53BP1 C-terminally fused to mApple fluorescent protein *Yang et al., 2015* |
| Transfected construct (human) | B_mCherry_IRES_neo3 | Addgene | 21044, RRID:Addgene_21044 | *Steigemann et al., 2009* |
| Transfected construct (human) | LeGO-Cer2 (Cerulean) | Addgene | 27338, RRID:Addgene_27338 | *Weber et al., 2008* |
| Transfected construct (human) | LeGO-V2 (Venus fluorescent protein) | Addgene | 27340, RRID:Addgene_27340 | *Weber et al., 2008* |
| Transfected construct (human) | LeGO-C2 (mCherry) | Addgene | 27339, RRID:Addgene_27339 | *Weber et al., 2008* |
| Antibody | Anti-p21 Waf1/Cip1 (Rabbit monoclonal) | Cell Signaling Technology | 2947, RRID:AB_330945 | Flow (1:200) |
| Antibody | Anti-p16 Ink4a (Mouse monoclonal) | Abcam | AB201980, RRID:AB_2891086 | Flow (1:200) |
| Antibody | Anti-gamma H2A.X (phospho S139) antibody (Rabbit monoclonal) | Cell Signaling Technology | 9718, RRID:AB_2118009 | Flow (1:200) |
| Antibody | Anti-PCNA (Mouse monoclonal) | Abcam | AB29, RRID:AB_303394 | IHC (1:500−1:2000) |

*Continued on next page*

*Continued*

| Reagent type (species) or resource | Designation | Source or reference | Identifiers | Additional information |
|---|---|---|---|---|
| Antibody | Alexa Fluor Plus 647 Secondary Antibody (Rabbit polyclonal) | Thermo Fisher Scientific | RRID:AB_A32733, RRID:AB_2633282 | Flow (1:500) |
| Antibody | BrdU-FITC (Mouse monoclonal) | BD Biosciences | 347583, RRID:AB_400327 | Flow (1:20) |
| Chemical compound, drug | Alexa Fluor 647 Phalloidin | Thermo Fisher Scientific | RRID:AB_A22287, RRID:AB_2620155 | IF (1:40,000) |
| Chemical compound, drug | Cisplatin | Hospira Australia | 88S035 | |
| Chemical compound, drug | Carboplatin | Abcam | ab120828 | |
| Chemical compound, drug | Olaparib | Selleck Chem | S1060 | |
| Chemical compound, drug | Palbociclib | Selleck Chem | S1116 | |
| Chemical compound, drug | RI-1 | Selleck Chem | S8077 | |
| Chemical compound, drug | PureLink RNase A | Thermo Fisher Scientific | 12091021 | |
| Chemical compound, drug | C12FDG (5-Dodecanoylamino fluorescein Di-β-D-Galactopyranoside) | Thermo Fisher Scientific | D2893 | |
| Chemical compound, drug | H33342 | Sigma | B2261 | 1 µg/ml |
| Chemical compound, drug | Propidium Iodide | Thermo Fisher Scientific | P3566 | |
| Chemical compound, drug | Thymidine | Selleck Chem | S4803 | |
| Chemical compound, drug | BrdU (5-Bromo-2′-Deoxyuridine) | Thermo Fisher Scientific | B23151 | |
| Commercial assay or kit | ImaGene Green C12FDG lacZ Gene Expression Kit | Molecular Probes | I2904 | |
| Other | Matrigel Basement Membrane | Bio-Strategy | BDAA354230 | |
| Software | Fiji/Image J | NIH | RRID:SCR_002285 | https://imagej.net/Fiji (*Schindelin et al., 2012*) |
| Software, algorithm | FlowJo | BD Biosciences | RRID:SCR_008520 | https://www.flowjo.com |
| Software, algorithm | GraphPad Prism (v9.1.0) | GraphPad | RRID:SCR_002798 | https://www.graphpad.com |
| Software, algorithm | Huygens Professional | Scientific Volume Imaging (SVI) | RRID:SCR_014237 | https://svi.nl/Huygens-Professional |
| Software, algorithm | LAS-X | Leica | RRID:SCR_013673 | https://www.leica-microsystems.com/products/microscope-software/p/leica-las-x-ls/ |

## Antibodies, plasmids, and reagents

The γH2A.X (S139) (AB26350), p16 (AB201980), and PCNA (AB29) antibodies were from Abcam (MA, USA), and p21 antibody (2947) was purchased from Cell Signaling Technology (MA, USA). BrdU-FITC antibody was purchased from BD-Biosciences (BD-347583). Alexa-647 Conjugated Phalloidin antibody was purchased from Thermo Fisher Scientific (A22287). The plasmids for FUCCI live-

cell imaging, mVenus-hGeminin(1/110) and mCherry-hCdt1(30/120), were a kind gift from Dr Atsushi Miyawaki (Riken, Japan). The LeGO plasmids were obtained from Addgene (#27338, #27339, #27340) (*Weber et al., 2008*). Thymidine (S4803), olaparib (S1060), palbociclib (S1116), and RI-1 (S8077) were from Selleck Chem (MA, USA). Deoxycytidine (sc-231247) was from Santa Cruz Biotechnology (TX, USA). BrdU was purchased from Thermo Fisher (B23151). Cisplatin was obtained from Hospira Australia (B23151) and carboplatin from Abcam (ab120828).

## Cell lines

The following LUAD cell lines were used: cell line (*Homo sapiens*) A549 ATCC CCL-185, RRID:CVCL_0023. Cell line (*H. sapiens*) NCI-H1299 ATCC CRL-5803, RRID:CVCL_0060. Cell line (*H. sapiens*) NCI-H1573 ATCC CRL-5877, RRID:CVCL_1478. All cell lines were authenticated by short tandem repeat polymorphism, single-nucleotide polymorphism, and fingerprint analyses, passaged for less than 6 months. All cell lines were confirmed as negative for mycoplasma contamination using the MycoAlert luminescence detection kit (Lonza, Switzerland).

Stable cell lines expressing the FUCCI biosensor were generated previously (*Hastings et al., 2020*). H2B-mCherry cells were generated by lentiviral transfection, followed by FACS sorting of low-expressing clones. Finally, dual chromobody and 53BP1 A549 cells were generated by lentiviral transfection with the PCNA chromobody, with low-expressing clones isolated by cell sorting. These were then subsequently transfected (lentiviral) with truncated form of 53BP1 fused to mApple (Apple-53BP1trunc), with cells sorted based on both mNeonGreen and mApple to isolate dual expressing clones.

All LUAD cell lines were cultured in Advanced RPMI (Gibco, 12633012) containing 1% FCS and 1% GlutaMAX (35050-061, Gibco) under standard tissue culture conditions (5% $CO_2$, 20% $O_2$) as previously described (*Hastings et al., 2020*; *Marini et al., 2018*).

## Colony formation assay and senescence-associated β-Gal assay

For colony formation assays, cells were seeded on 6-well plates, pulsed with cisplatin (or not), and 1–2 weeks later, colonies were stained with 0.5% crystal violet and counted using ImageJ/Fiji software. For β-Gal assays, cells were seeded on 6-well plates, pulsed with cisplatin (or not), fixed and stained at 3 days, following manufacturer's protocol (Cell Signaling Technology, #9860). Unpaired Student's t-tests along with bar graphs were generated using GraphPad Prism (v9.1.0).

## LeGO clonal analysis

A549, NCI-H1573, and NCI-H1299 cells were transfected with LeGO lentiviral particles (Addgene plasmids #27338, #27339, #27340) *Weber et al., 2008* following the method described in *Weber et al., 2012*. Cells were treated with/without cisplatin, and images were taken at 3 days and at 21 days (A549 and NCI-H1299) or 42 days (NCI-H1573) after cisplatin exposure. 100 images were taken per timepoint and per condition (three replicates), and the experiment was done twice. The total number of clones (unique colour cues) and the number of cells within each clone was determined. Briefly, images are opened and converted to 16-bit .tif files. An image is duplicated and converted to RGB overlay. The duplicate has background subtracted using a rolling ball at 250 considering colours separately and using a sliding parabaloid. The image is smoothed using a mean filter radius 5. Using the 'find maxima' function, a point within individual cells is identified and then enlarged to a circle radius of 5 pixels. These regions of interest (ROI) are then applied as a mask to the unprocessed, raw, image data and the average red, green, and blue values within these ROI collected and exported in .csv format. RGB values from each of the .csv files for each of the 100 images are compiled. Data from cells where an R, G, or B value is too high or too low are removed. 512 unique colours were identified, and cells were classified and assigned to each of the 512 colours. More than 90% of all cells were assigned to one of the 64 most represented colours, with analysis performed using these 64 groups and positive clonal colour assigned when the colour represented >0.1% of the population. Graphs were generated using GraphPad Prism (v9.1.0).

## Animal experiments

Animal experiments were conducted in accordance with the Garvan/St Vincent's Animal Ethics Committee (guidelines ARA 18_17, ARA_16_13) and in compliance with the Australian Code of Practice

for Care and Use of Animals for Scientific Purposes. Mice were kept in standard housing at a 12 hr daylight cycle and fed ad libitum. Cage enrichment refinement was undertaken with mice implanted with mammary optical imaging windows, supplying the fully plastic IVC cages with papier-mâché domes, feeding supplied in trays on the cage floor and soft tissues as nesting material. For in vivo xenograft models, A549 cells ($2 \times 10^6$) were resuspended in 100 µl PBS:Matrigel (1:1) and injected subcutaneously into the flanks of BALB/c-Fox1nuAusb mice (Australian BioResource). Tumour growth was assessed twice weekly by calliper measurement, and mice were randomised to treatment arms when tumours reached 150 mm$^3$ (using the formula: width$^2 \times$ length $\times$ 0.5). Carboplatin (60 mg/kg) was delivered by a single i.p injection. Tumours were harvest at 3–7 days post treatment and analysed by IHC for cell size and PCNA-positive staining.

## Implantation of optical imaging windows

BALB/c-Foxn1nu/Ausb mice were injected with $1 \times 10^6$ A549-FUCCI subcutaneously near the inguinal mammary fat pad. Following the development of palpable tumours, mice were engrafted with titanium mammary imaging windows (Russell Symes & Company) as described previously (*Gligorijevic et al., 2009*; *Kedrin et al., 2008*; *Nobis et al., 2017*; *Ritsma et al., 2013*). Briefly, mice were treated with 5 mg/kg of the analgesic carprofen (Rimadyl) in pH neutral drinking water 24 hr prior and up to a minimum of 72 hr post surgery. Mice further received subcutaneous injections of buprenorphine (0.075 mg/kg, Temgesic) immediately prior to and 6 hr post surgery. The titanium window was prepared 24 hr prior to surgery by gluing a 12 mm glass coverslip (Electron Microscopy Science) using cyanoacrylate to the groove on the outer rim of the titanium window. Following anaesthetic induction at 4% isoflurane delivered via a vaporizer (VetFlo) supplemented with oxygen, mice were kept at a steady 1–2% maintenance anaesthesia for the duration of the surgery on a heated pad. The incision site was disinfected using 0.5% chlorhexidine/70% ethanol. A straight incision was made into the skin above the developed subcutaneous tumour and following blunt dissection of the skin surrounding the incision a purse string suture (5-0 Mersilk, Ethicon) placed. The windows were then inserted and held in place by tightening the suture, disappearing along with the skin into the groove of the window and tied off. Mice were allowed to recover for a minimum of 72 hr post surgery, actively foraging, feeding, and grooming within minutes from being removed from the anaesthesia respirator. A minimum of 24 hr prior to imaging and treatment mice were weaned off the carprofen analgesic in the drinking water.

## In vivo imaging

Mice were imaged under 1–2% isofluorane on a heated stage (Digital Pixel, UK) prior to and 1 day, 2 days, 3 days, and 7 days after i.p. injection of 60 mg/kg carboplatinum (Sigma) or the saline vehicle. Multi-photon imaging was performed using a Leica DMI 6000 SP8 confocal microscope using a 25 $\times$ 0.95 NA water immersion objective on an inverted stage. For A549-FUCCI imaging the Ti:Sapphire femto-second laser (Coherent Chameleon Ultra II, Coherent) excitation source operating at 80 MHz was tuned to 920 nm and the RLD-HyD detectors with 460/40, 525/50, and 585/40 bandpass emission filters used to detect the second harmonic generation (SHG) of the collagen I, mAzamiGreen, and mKO2, respectively. Images were acquired at a line rate of 400 Hz, 512 $\times$ 512 pixel, and a line average of 8.

## Flow cytometry analysis and sorting

Samples for flow cytometry were fixed in $-20°$C ethanol overnight, and then stained with a primary antibody against p21 (Cell Signal Technology, 2947), p16 (Abcam, ab201980), or gamma-H2A.X (S139) (Abcam, ab26350) followed by incubation with an Alexa Fluor 647 secondary antibody (Invitrogen). For DNA content analysis, cells were stained with 1µg/ml propidium iodide (PI) and treated with 0.5mg/ml RNAaseA for at lease 1 h prior to analysis. Flow cytometry was performed using a Beckman CytoFlex S. For BrdU incorporation analysis, cells were incubated with BrdU at 10 µM for 2 hr before overnight ethanol fixation at $-20°$C. An antibody against BrdU coupled with FITC (BD-347583) was used for staining, and flow cytometry was done using a Beckman CytoFlex S. For senescence assays, we used ImaGene Green C12FDG lacZ Gene Expression Kit (Molecular Probes, I-2904). Three days after cisplatin exposure, cells were incubated for 30 min with Bafilomycin A1 (Sigma, B1793) in RPMI medium without phenol red (Gibco) supplemented with 1% FBS before

adding $C_{12}FDG$ to the media at 20 µM final concentration. Cells were incubated for 60 min prior to 15 min fixation with 4% PFA and processed for FACS analysis. Flow cytometry was performed using a Beckman CytoFlex S.

For cell sorting and RNAseq analysis, A549 cells were treated with or without cisplatin (5 µg/ml) for 2 hr, and then allowed to recover for 3 days. Cells were collected by trypsinisation and then sorted using a BD FACS Aria III. The gates for stable and enlarged cells were determined by running untreated control cells and identifying cell size based on FSC and SSC (FSC vs. SSC) area parameters. Doublets were excluded based on area and height parameters of FSC and SSC. Sorted cells were frozen as a pellet in dry ice and stored at −80°C until RNA purification.

## Immunofluorescence and live-cell imaging

Cells were grown on Histogrip (Life Technologies)-coated glass coverslips and fixed with 3.7% formaldehyde diluted in PHEM buffer (60 mM pipes, 25 mM HEPES, 1 mM EGTA, 2 mM $MgCl_2$) with 0.5% Triton X-100 for 10 min. All cells were washed and then blocked (3% BSA, 0.1% Tween 20 in PBS) for 30 min. Cells were incubated with primary antibodies for 2 hr at room temperature in blocking solution. DNA was stained with H33342 and imaged using an EVOS FL2 Auto Imager (Thermo Fisher) or a Leica SP8-X confocal with white light laser using either a 20× (NA 0.75) or 63× (NA 1.40) objective. In some cases, 0.3 µm Z-sections were taken and displayed as 2D slices or maximum projections using Fiji (Image J v2.1.0/1.53c) and compiled using Adobe Photoshop CC 2020 software. Deconvolution and 3D volume renderings were performed using Huygens Professional Software (Scientific Volume Imaging, v20.04), while nuclear size analysis was performed using StarDist (*Schmidt et al., 2018*) plugins for Fiji/ImageJ. Live-cell imaging and IncuCyte (Sartorius) proliferation assays were performed as previously described (*Hastings et al., 2020*; *Rogers et al., 2018*). Briefly, for live-cell imaging, cells were seeded at 35% confluence on 6- or 12-well plates and imaged using a Leica DMI6000 with a 20× NA 0.4 objective. Images were taken every 10–20 min for up to 72 hr. Individual cells were followed and scored for nuclear envelope breakdown (NEBD) and first signs of anaphase as previously described (*Caldon and Burgess, 2019*). Mitotic length = NEBD to anaphase, while interphase length = anaphase to next daughter cell NEBD. Only the first daughter cell to divide was followed and annotated. For IncuCyte assays, cells were seeded on 12- or 24-well plates and filmed for up to 4 days at 4 hr intervals. Confluence and nuclear masks were generated and used to determine cell proliferation as previously described (*McCloy et al., 2014*). Statistical analysis, along with box and violin plots, was generated using GraphPad Prism (v9.1.0).

For 53BP1 and PCNA chromobody experiments, cells were seeded on 8-Well Ibidi Polymer Coverslip µ-Slides (#80826), synchronised with thymidine or palbociclib and pulsed with cisplatin for 2 hr, before imaging on a Lecia SP8 confocal microscope fitted with a white light laser, hybrid detectors (HyD), a 63X HC PL APO CS2 (NA 1.40) objective and stage top incubator system set at 37°C and 5% $CO_2$. Multiple X/Y positions and a 10 µm z-stack (1 µm Z-section) were taken every 30 min for 72 hr, with 4D deconvolution and volume rendering performed with Huygens Professional (v20.04) software (Netherlands). 53BP1 and PCNA foci analysis was performed on 2D maximim intensity projections using appropriate thresholds coupled with the Analyse Particles module within ImageJ/Fiji. The pattern of PCNA foci was used to position cells in early, mid, or late S phase, as previously described (*Burgess et al., 2012*; *Charrasse et al., 2017*).

## Immunohistochemistry

Immunohistochemistry was performed on formalin-fixed paraffin-embedded sections using the Leica BOND RX (Leica, Wetzlar, Germany). Slides were first dewaxed and rehydrated, followed by heat-induced antigen retrieval performed with Epitope Retrieval Solution 1 BOND (Leica). PCNA primary antibody was diluted 1:500 (Abcam, ab29) in Leica antibody diluent and incubated for 60 min on slides. Antibody staining was completed using the Bond Polymer Refine IHC protocol and reagents (Leica). Slides were counterstained on the Leica Autostainer XL (Leica). Leica CV5030 Glass Coverslipper (Leica) and brightfield images were taken on the Aperio CS2 Slide Scanner (Leica). Quantification of PCNA staining was performed on three fields of view for each tumour section using QuPath (v0.2.3)(*Bankhead et al., 2017*). Student's t-test statistical analysis, along with dot plots and bar graphs, was generated using GraphPad Prism (v9.1.0).

## RNA isolation, RNA sequencing (RNAseq), SNV alignment and analysis

Cell pellets were obtained from the different conditions/populations. Cell pellets were frozen in dry ice prior to storage at −80°C. Total RNA was purified using miRNeasy Micro Kit (QIAGEN, 217084) following the manufacturer's protocol, including a DNase treatment. RNA concentration and quality were also measured by Qubit and Nanodrop. Samples were only used if they showed a 260/280 ratio >2.0 (Nanodrop). RNA integrity was determined on an Agilent 2100 Bioanalyser, and samples were only used if they showed a RNA integrity number (RIN) of >8. Three sets of RNA were collected per condition. Compliant samples were sent to the Australian Genome Research Facility (AGRF) for RNA sequencing with poly(A) selection. Briefly, 20 million 100 bp single-end RNAseq was conducted on an Illumina NovaSeq platform. The library was prepared using the TruSeq stranded RNA sample preparation protocol (Illumina).

The cleaned sequence reads were aligned against the *H. sapiens* genome (Build version hg38), and the RNAseq aligner, 'Spliced Transcripts Alignment to a Reference (STAR)' aligner (v2.5.3a) (*Dobin et al., 2013*), was used to map reads to the genomic sequence. Transcripts were assembled using the StringTie tool v1.3.3 (*Pertea et al., 2015*) with the read alignment (hg38) and reference annotation-based assembly option. Raw data were deposited in the NCBI Gene Expression Omnibus data repository accession number GSE161800.

The raw data from each cell line was aligned to the human genome reference build GRCh38/hg38 using STAR aligner v2.5.3a by AGRF. Single-nucleotide variations (SNVs) were identified using SNV caller Freebayes (v1.3.1; https://github.com/ekg/freebayes *Garrison et al., 2021* (copy archived at swh:1:rev:60850dc518fc453622cbb40ad6dd9f67644ed859); *Gonzalez, 2021*), and annotated using Bcftools (v1.9) (*Danecek and McCarthy, 2017*) with database NCBI dbSNP (v146) (*Sherry et al., 2001*). Heatmaps, principal component analysis and biological coefficient variant plots were made using R language and software (The R Foundation) with the DESeq2 package (*Love et al., 2014*). The log2 (fold change) scale was normalised and transformed by considering library size or other normalisation factors. The transformation method and the variance stabilising transformation (VST) (*Anders and Huber, 2010*) for over-dispersed counts have been applied in DESeq2. The VST is effective at stabilising variance because it considers the differences in size factors, such as the datasets with large variation in sequencing depth (*Love et al., 2014*). Canonical pathway analysis of known proliferation, cell cycle, migration, and cell death-related signalling pathways was conducted using the Ingenuity Pathway Analysis software (QIAGEN), as previously described (*Johnson et al., 2020*). Briefly, minimum significance cut-offs of p-value>0.05 and Z-scores of >2 and <−2 were applied for pathways analysis. For GSEA, a ranked gene list was prepared from proliferative versus arrest and analysed with GSEA 4.1.0 software (https://www.gsea-msigdb.org/gsea/index.jsp) using a curated gene set of canonical pathways (2868 gene sets) (https://www.gsea-msigdb.org/gsea/msigdb/collections.jsp#C2) (*Mootha et al., 2003*; *Subramanian et al., 2005*). The enrichment map was generated using Cytoscape 3.8.2 software (https://cytoscape.org/) (*Shannon et al., 2003*), using p-value (<0.005) and false discovery rate (FDR) (q < 0.1) cut-offs. Volcano and dot plots were generated using GraphPad Prism (v9.1.0) and figures compiled using Adobe Illustrator Creative Cloud (v25).

## Acknowledgements

The authors would like to acknowledge Dr Atsushi Miyawaki (Riken, Japan) for provision of the mVenus-hGeminin(1/110) and mKO-hCdt1(30/120) constructs for FUCCI imaging. Dr Liz Caldon for her insightful and helpful comments. The Patricia Helen Guest Fellowship, Doherty Swinhoe Family Foundation for their generous support. The authors acknowledge the ANZAC Microscopy and Flow Facility, the Sydney Informatics Hub, and the use of the University of Sydney's high-performance computing cluster, Artemis.

# Additional information

## Funding

| Funder | Grant reference number | Author |
|--------|------------------------|--------|
| National Breast Cancer Foundation | IIRS-18-103 | Andrew Burgess |
| Tour de Cure | RSP-230-2020 | Andrew Burgess |
| Cancer Institute NSW | 10/FRL/3-02 | Andrew Burgess |
| Cancer Institute NSW | 2013/FRL102 | David R Croucher |
| Cancer Institute NSW | 15/REG/1-17 | David R Croucher |

The funders had no role in study design, data collection and interpretation, or the decision to submit the work for publication.

## Author contributions

Alvaro Gonzalez Rajal, Conceptualization, Data curation, Formal analysis, Validation, Investigation, Visualization, Methodology, Writing - review and editing; Kamila A Marzec, Rachael A McCloy, Formal analysis, Investigation; Max Nobis, Venessa Chin, Jordan F Hastings, Investigation, Methodology; Kaitao Lai, Data curation, Formal analysis, Visualization, Methodology; Marina Kennerson, Supervision, Project administration; William E Hughes, Formal analysis, Methodology; Vijesh Vaghjiani, Investigation; Paul Timpson, Supervision, Methodology; Jason E Cain, Formal analysis, Investigation, Writing - review and editing; D Neil Watkins, Conceptualization, Data curation, Supervision, Investigation, Project administration, Writing - review and editing; David R Croucher, Conceptualization, Data curation, Supervision, Funding acquisition, Project administration, Writing - review and editing; Andrew Burgess, Conceptualization, Data curation, Formal analysis, Supervision, Funding acquisition, Validation, Investigation, Visualization, Methodology, Writing - original draft, Project administration, Writing - review and editing

## Author ORCIDs

Alvaro Gonzalez Rajal ![ORCID] https://orcid.org/0000-0002-9230-0339
Kamila A Marzec ![ORCID] https://orcid.org/0000-0002-3051-2205
Max Nobis ![ORCID] http://orcid.org/0000-0002-1861-1390
Jordan F Hastings ![ORCID] https://orcid.org/0000-0003-2839-4358
Kaitao Lai ![ORCID] https://orcid.org/0000-0002-9420-9352
Marina Kennerson ![ORCID] https://orcid.org/0000-0003-3332-5074
Jason E Cain ![ORCID] https://orcid.org/0000-0003-3987-5894
D Neil Watkins ![ORCID] https://orcid.org/0000-0001-8218-4920
David R Croucher ![ORCID] https://orcid.org/0000-0003-4965-8674
Andrew Burgess ![ORCID] https://orcid.org/0000-0003-4536-9226

## Ethics

Animal experimentation: All experiments were carried out in compliance with the Australian code for the care and use of animals for scientific purposes and in compliance with Garvan Institute of Medical Research/St. Vincent's Hospital Animal Ethics Committee guidelines (ARA_18_17, ARA_16_13).

## Decision letter and Author response

Decision letter https://doi.org/10.7554/eLife.65234.sa1
Author response https://doi.org/10.7554/eLife.65234.sa2

# Additional files

## Supplementary files

- Supplementary file 1. Summary table of comparative RNAseq data for pre-sorted untreated A549 control cells verse cisplatin pulsed cells.
- Supplementary file 2. Summary table of comparative RNAseq data for pre-sorted untreated A549 control cells verse post-sorted enlarged cisplatin pulsed cells.
- Supplementary file 3. Summary table of comparative RNAseq data for pre-sorted untreated A549 control cells verse post-sorted stable size cisplatin pulsed cells.
- Supplementary file 4. Summary table of comparative RNAseq data for post-sorted enlarged cisplatin pulsed cells A549 cells verse post-sorted stable size cisplatin pulsed cells.
- Transparent reporting form

## Data availability

Raw RNAseq data has been uploaded to the NCBI Gene Expression Omnibus (GEO) data repository with the accession number GSE161800.

The following dataset was generated:

| Author(s) | Year | Dataset title | Dataset URL | Database and Identifier |
|---|---|---|---|---|
| Burgess A, Gonzalez Rajal A, Lai K | 2020 | Pulsed cisplatin treatment of A549 lung adenocarcinoma cells | https://www.ncbi.nlm.nih.gov/geo/query/acc.cgi?acc=GSE161800 | NCBI Gene Expression Omnibus, GSE161800 |

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
