## [Decision Letter]

**Acceptance summary:**

In this manuscript the authors use biosensor imaging and single-cell fate tracking and discover a non-genetic mechanism of resistance that drives recovery and regrowth in a subset of lung adenocarcinoma cell lines with varying p53 status. The authors show that early S phase cells have greater resistance to cisplatin due to their increased capacity to repair double stranded breaks. They then show that combined treatment of cisplatin and a PARP inhibitor reduces resistance to cisplatin in lung cancer cells.

**Decision letter after peer review:**

Thank you for submitting your article "Resistance to platinum chemotherapy in lung adenocarcinoma is driven by a non-genetic, cell-cycle dependent mechanism" for consideration by *eLife*. Your article has been reviewed by 3 peer reviewers, and the evaluation has been overseen by Maureen Murphy as the Senior and Reviewing Editor. The following individual involved in review of your submission has agreed to reveal their identity: Eric Batchelor (Reviewer #2).

The reviewers have discussed the reviews with one another and the Reviewing Editor has drafted this decision to help you prepare a revised submission.

Summary:

In this manuscript the authors explore a non-genetic mechanism of resistance that drives recovery and regrowth in a subset of lung adenocarcinoma cell lines with varying p53 status. The authors show that early S phase cells have greater resistance to cisplatin due to their increased capacity to repair double stranded breaks. Their findings demonstrate that combined treatment of cisplatin and a PARP inhibitor reduced resistance to cisplatin in A549 cells. The attempts to identify a non-genetic resistance mechanism to cisplatin in lung adenocarcinoma cell lines are interesting, relevant and compelling. The findings after extensive biosensor imaging and single-cell fate tracking suggest a cell-cycle dependent role in cisplatin resistance. Further development of these findings could one day help improve the timing of combination chemotherapies for greater cancer treatment efficacy and reduction of fractional cell killing. The paradigms established are likely to be of interest to the broad *eLife* audience, pending revisions. Many of the requested revisions deal with increased experimental and statistical clarity; there is also concern that the relevance of p53 status is initially addressed but that this question then loses focus, thus detracting from broad overall relevance of the findings.

Essential revisions:

1. In Figure 2D, for the analysis of clonal outgrowth, I suggest quantifying the number of clones, perhaps by normalizing number of colors by number of cells for each of the lines. It looks like the NCI-H1573 p53248L cell line has a lower level of clonal variation than the other two cell lines, being dominated to a greater extent by a single clone. Does this hold true statistically with repeats of the experiment? Given the potential dependence on p53 status for the in vivo results in Figure 3, I would suggest performing the same in vivo analysis for NCI-H1573 cells as was done for the other two cell lines shown in Figure 3 to see what happens with a gain of function mutant form of p53.

2. The experiment in Figure 5 Supp 1, tracking the cell cycle in response to cisplatin without the use of a synchronization technique, is a cleaner experiment than that shown in Figure 5. Synchronization methods may perturb the general cell state in unforeseen ways – simply measuring the cell cycle stage without the need to enrich for a specific population already appears to be providing convincing evidence. I would recommend increasing the numbers of cells analyzed to improve statistical significance and moving those results to the main figure.

3. PARP inhibition potentially has many pleiotropic effects beyond inhibition of HR. It would be cleaner to validate their findings by looking at the cell cycle dynamics in response to cisplatin upon knockdown of HR-associated proteins, such as BRCA1/2. Similarly, given the off-target effects of PARP inhibitors, it would be useful to test other PARP inhibitors such as rucaparib.

4. The manuscript starts with a comparison between three different LUAD cell lines with noted different p53 statuses; however, by the end of the manuscript, the studies are only performed on p53 WT A549 cells. Given the importance of p53 status in lung cancer, and the fact that a large number of cancer cells are not wild-type for p53, interest in the findings would be improved if the connection between cell cycle stage at time of cisplatin treatment and the decision between proliferation / senescence were shown to be maintained in p53-mutant and p53-null cells as well, or alternatively could the authors show how the response is altered.

5. The authors show that a cell population that remains proliferative in response to an initial dose of cisplatin can, after a long time, generate a similar distribution of proliferative vs senescent cells in response to a second dose of cisplatin (Figure 1). This fits nicely with the model of cell cycle dependence on the proliferation / senescence bifurcation, as desynchronized cell cycle distribution would be re-established at some point following recovery from the initial cisplatin treatment. The model presented in this work would be further validated by considering different time points after the first cisplatin treatment, quantifying the cell cycle distribution, and then challenging cells with the second dose of cisplatin and determining the number of proliferative cells. This could potentially help inform better optimized cisplatin dosing regimens to decrease the percentage of cells that escape treatment.

6. It would be interesting, if possible, to analyze the transcriptome of cells in the different cell-cycle stage (G1, early S, late S, G2) either in sorted cells or in synchronized cells, in order to see whether they recapitulate the different transcriptome observed in cells proliferating or not proliferating after cisplatin pulse. If this is possible that would be ideal but if impossible, perhaps the authors can mention this in the Discussion.

---

## [Author Response]

Essential revisions:1. In Figure 2D, for the analysis of clonal outgrowth, I suggest quantifying the number of clones, perhaps by normalizing number of colors by number of cells for each of the lines. It looks like the NCI-H1573 p53248L cell line has a lower level of clonal variation than the other two cell lines, being dominated to a greater extent by a single clone. Does this hold true statistically with repeats of the experiment?

We have now included additional biological repeats (n=3) for the LeGo data, and also presented normalised summary data, which as noted confirms that there is a slightly lower starting level of colour complexity in NCI-H1573 cells. This data is presented in the new Figure 2 Supplement 1D.

Given the potential dependence on p53 status for the in vivo results in Figure 3, I would suggest performing the same in vivo analysis for NCI-H1573 cells as was done for the other two cell lines shown in Figure 3 to see what happens with a gain of function mutant form of p53.

As suggested we have now performed the in vivo analysis for the NCI-H1573 cell line. This data is now presented in Figure 3. In addition, we have also added addition in vivo data both NCI-H1299 and NCI-1573 FUCCI cell lines using the optical window assays in Figure 5 Supplement 3A-B.

2. The experiment in Figure 5 Supp 1, tracking the cell cycle in response to cisplatin without the use of a synchronization technique, is a cleaner experiment than that shown in Figure 5. Synchronization methods may perturb the general cell state in unforeseen ways – simply measuring the cell cycle stage without the need to enrich for a specific population already appears to be providing convincing evidence. I would recommend increasing the numbers of cells analyzed to improve statistical significance and moving those results to the main figure.

As requested, we have now significantly re-organised Figure 5 to focus solely on the asynchronous data. In addition, we have added 3 new supplements for Figure 5. In these we provide additional analysis from 5 independent biological repeats of asynchronously cisplatin treated cells, analysed by 3 different researchers, covering more than 400 cells to improve the statistical significance of this data.

3. PARP inhibition potentially has many pleiotropic effects beyond inhibition of HR. It would be cleaner to validate their findings by looking at the cell cycle dynamics in response to cisplatin upon knockdown of HR-associated proteins, such as BRCA1/2. Similarly, given the off-target effects of PARP inhibitors, it would be useful to test other PARP inhibitors such as rucaparib.

We have now included additional data where we have targeted homologous recombination using the RAD51 inhibitor RI-1. This data is now included as Figure 8 Supplement 1. Importantly, this data supports the results we observed with the PARP inhibitor, indicating that inhibition of DNA repair pathways can reduce the likelihood that cells are able to sufficiently repair damage during the first cell cycle, thereby limiting proliferative capacity of cells and promoting G2-exit.

4. The manuscript starts with a comparison between three different LUAD cell lines with noted different p53 statuses; however, by the end of the manuscript, the studies are only performed on p53 WT A549 cells. Given the importance of p53 status in lung cancer, and the fact that a large number of cancer cells are not wild-type for p53, interest in the findings would be improved if the connection between cell cycle stage at time of cisplatin treatment and the decision between proliferation / senescence were shown to be maintained in p53-mutant and p53-null cells as well, or alternatively could the authors show how the response is altered.

We agree with this point, and consequently we have now added significantly more data to help elucidate the links between p53 status and cell fate outcomes. This includes addition of optical window based in vivo data for NCI-H1573 (p53 mutant) and NCI-H1299 (p53 null) cells. Additional cell fate mapping of FUCCI expressing NCI-H1299 cells, which lack p53, results in more cells undergoing multiple divisions. In our previous publication, we demonstrated that loss of p53 correlates with a failure to upregulate p21 (Figure 1D, (Hastings et al., 2020)), which correlated with much lower percentage of cells undergoing a G2-exit. The lack of strong p21 expression and compromised cell cycle checkpoint response, likely contributes to the overall higher rates of proliferation in p53 null cells. Interestingly, we still observed a trend towards more divisions in NCI-H1299 cells that were exposed during late G1 and early S phase. In addition, we have added additional analysis the p53 siRNA knockdown data from the original paper (Hastings et al., 2020) to show that knockdown of p53 increases the number of divisions in cisplatin treated cells. Taken together, these data suggest that the cell cycle specific resistance we report here is not dependent on p53, however, loss of p53/p21 reduces the ability of highly damaged cells undergo cell cycle arrest and exit, leading to a higher percentage of cells undergoing aberrant mitotic divisions.

5. The authors show that a cell population that remains proliferative in response to an initial dose of cisplatin can, after a long time, generate a similar distribution of proliferative vs senescent cells in response to a second dose of cisplatin (Figure 1). This fits nicely with the model of cell cycle dependence on the proliferation / senescence bifurcation, as desynchronized cell cycle distribution would be re-established at some point following recovery from the initial cisplatin treatment. The model presented in this work would be further validated by considering different time points after the first cisplatin treatment, quantifying the cell cycle distribution, and then challenging cells with the second dose of cisplatin and determining the number of proliferative cells. This could potentially help inform better optimized cisplatin dosing regimens to decrease the percentage of cells that escape treatment.

We have now added additional flow cytometry data to provide greater temporal clarity between initial treatment and day 21 of cisplatin recovery in A549 cells. This data is now included in Figure 1 Supplement 1C.

6. It would be interesting, if possible, to analyze the transcriptome of cells in the different cell-cycle stage (G1, early S, late S, G2) either in sorted cells or in synchronized cells, in order to see whether they recapitulate the different transcriptome observed in cells proliferating or not proliferating after cisplatin pulse. If this is possible that would be ideal but if impossible, perhaps the authors can mention this in the Discussion.

We agree that this would be an interesting experiment, unfortunately it is beyond the scope of this paper. We have included this as a possibility for future directions in the discussion.